# PiCO: Peer Review in LLMs based on the Consistency Optimization

## Abstract

Existing large language models (LLMs) evaluation methods typically focus on testing the performance on some closed-environment and domain-specific benchmarks with human annotations. In this paper, we explore a novel **unsupervised evaluation direction**, utilizing *peer-review* mechanisms to measure LLMs automatically without any human feedback. In this setting, both open-source and closed-source LLMs lie in the same environment, capable of answering unlabeled questions and evaluating each other, where each LLM's response score is jointly determined by other anonymous ones. To obtain the ability hierarchy among these models, we assign each LLM a learnable capability parameter to adjust the final ranking. We formalize it as a constrained optimization problem, intending to maximize the consistency of each LLM's capabilities and scores. The key assumption behind is that high-level LLM can evaluate others' answers more accurately than low-level ones, while higher-level LLM can also achieve higher response scores. Moreover, we propose three metrics called PEN, CIN, and LIS to evaluate the gap in aligning human rankings. We perform experiments on multiple datasets with these metrics, validating the effectiveness of the proposed approach.

## 1 Introduction

> Goodhart's Law: *"When a measure becomes a target, it ceases to be a good measure."*

Large language models (LLMs)[11, 2, 12, 43] have achieved remarkable success across a variety of real-world applications [54, 32, 36, 52]. With the increasingly widespread application of these models, there is an urgent need for an effective evaluation method to ensure that their performance and usability meet the growing demands. To assess the ability level of LLMs, a large number of evaluation benchmarks have been proposed by using some small and domain-specific datasets with human-curated labels, such as MMLU [26], HELM [30], Big-Bench[39], GLUE[45]. However, these benchmarks can only measure LLMs' core capability on a confined set of tasks (e.g. multi-choice knowledge or retrieval questions), which fails to assess their alignment with human preference in open-ended tasks adequately [16, 28, 34]. On the other hand, these evaluations may suffer from *benchmark leakage* issue, referring that the evaluation data is unknowingly used for model training, which can also lead to misleading evaluations [49, 56]. Therefore, blindly improving scores on these public benchmarks cannot always yield a large language model that truly satisfies human requirements.

For assessing human preferences, recent studies have focused on building crowdsourced battle platforms with human ratings as the primary evaluation metric. Typical platforms include Chatbot Arena [55], MT-Bench [55], and AlpacaEval [29]. It constructs anonymous battles between chatbots in real-world scenarios, where users engage in conversations with two chatbots at the same time and rate their responses based on personal preferences. While human evaluation is the gold standard for

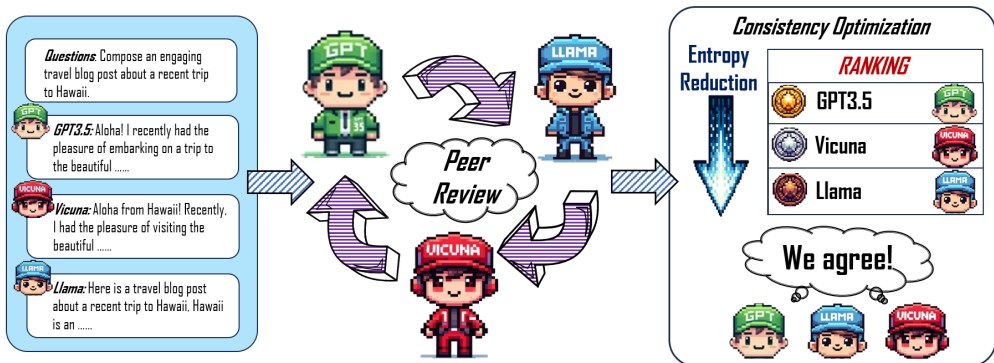

Figure 1: The framework of PiCO. In this framework, both open-source and closed-source LLMs lie in the same environment, capable of answering unlabeled questions and evaluating each other, where each LLM's response score is jointly determined by other anonymous ones. We assign each LLM a learnable capability weight to optimize the score ranking based on the *consistency assumption*, while reducing the entropy of the *peer-review* evaluation system. The consistency optimization aims to find a final score ranking that all LLMs "agree" it.

measuring human preferences, it is exceptionally slow and costly[55]. In addition, adding a new LLM to the crowdsourced battle platforms also poses a cold-start issue [15]. Thus, a fundamental question arises: *can we construct an unsupervised LLMs evaluation system without relying on any human feedback*?

Actually, in real human evaluation systems, people build their ability hierarchy based on different empirical assumptions. For example, majority voting [22, 10, 40] and rating voting [5] methods are widely used during the decision-making process, which are based on the wisdom of the crowds [40, 13, 50] and have been proven to lead to better results than that of an individual. Moreover, in the established practice of *peer-review* in academic research, scholars evaluate their academic level rankings based on the *consistency assumption*, *i.e.*, scholars with stronger abilities have stronger persuasiveness for evaluating others, and can also obtain higher achievements. This paper attempts to explore whether similar phenomena exist in the LLMs evaluation systems.

In this work, we propose **PiCO**, a **P**eer review approach **i**n LLMs based on **C**onsistency **O**ptimization. In this setting, LLMs themselves act as "reviewers", engaging in mutual assessments to achieve comprehensive, efficient, and performance evaluations without relying on manually annotated data. This method aims to address the limitations of existing evaluation approaches and provide insights into LLMs' real-world capabilities. As shown in Figure 1, both open-source and closed-source LLMs lie in the same environment and answer the open-ended questions from an unlabeled dataset. Then, we construct anonymous answer pairs, while randomly selecting other LLMs as "reviewers" to evaluate both responses with a learnable confidence weight $w$. Finally, we employ this weight and calculate the response scores $G$ for each LLM based on the weighted joint evaluation. It is worth noting that the whole *peer-review* process works in an unsupervised way, and our goal is to optimize the confidence weights that re-rank the LLMs to be closer to human rankings.

To achieve this, we formalize it as a constrained optimization based on the consistency assumption. We maximize the consistency of each LLM's capability $w$ and score $G$ while adjusting the final ranking to align with human preference more closely. The key assumption behind this is that high-level LLM can evaluate others' answers more accurately (confidence) than low-level ones, while higher-level LLM can also achieve higher answer-ranking scores. As a result, the entropy (controversy) of the whole *peer-review* evaluation system can be minimized. In other words, the consistency optimization aims to find a final score ranking that all LLMs have no "disputes" regarding.

To evaluate the gap in aligning human rankings, we propose three metrics called PEN (**P**ermutation **En**tropy), CIN (**C**ount **In**versions), LIS (**L**ongest **I**ncreasing **S**ubsequence). The experiments are conducted on multiple crowdsourcing datasets and validated on these three metrics. The experimental results demonstrate that the proposed PiCO framework can effectively obtain a large language models' leaderboard closer to human preferences.

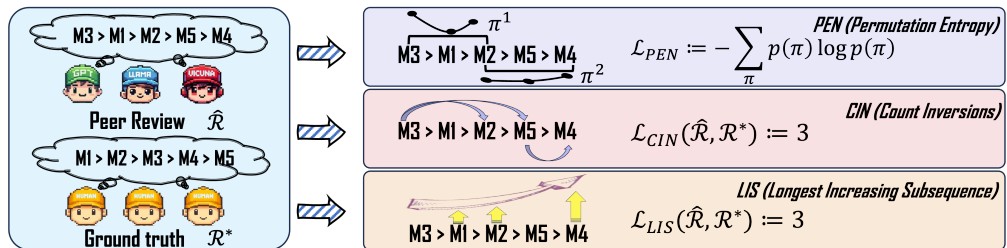

Figure 2: Preference alignment metric. Three metrics for evaluating the gap with human preferences called PEN, CIN, and LIS, respectively

The contributions of this paper can be summarized as follows.

- We explore a novel unsupervised LLM evaluation direction without human feedback, utilizing *peer-review* mechanisms to measure LLMs automatically. All LLMs can answer unlabeled questions and evaluate each other.
- A constrained optimization based on the consistency assumption is proposed to re-rank the LLMs to be closer to human rankings.
- We propose three metrics called PEN, CIN, and LIS on the PiCO framework for evaluating the gap with human preferences.
- The experiments with these metrics on three crowdsourcing datasets validate the effectiveness of the proposed approach.

## 2 The Proposed Approach

In this section, we first describe the problem definition and preference alignment evaluation, and then introduce the proposed PiCO framework in detail.

### 2.1 Definition and Metrics

**Problem Definition.** In this subsection, we aim to measure the ability level of LLMs automatically without relying on human annotations. Thus we consider an unsupervised LLM evaluation scenario with an unlabeled dataset $\mathcal{Q}$ consisting of $n$ open-ended questions, where $\mathcal{Q} = \{Q_i\}_{i=1}^{n}$. In addition, we have a large language model pool $\mathcal{M} = \{M_j\}_{j=1}^{m}$, which includes both open-source and closed-source models. Write $M_1 \succ M_2$ to indicate that the LLM $M_1$ has stronger capabilities than the LLM $M_2$. Thus, we can assume that the ground-truth ranking $\mathcal{R}^*$ alignment with human preferences,

$$\mathcal{R}^* := [M_1 \succ M_2 \succ M_3 \succ ... \succ M_m], \tag{1}$$

and assume that the learned ranking $\hat{\mathcal{R}}$ by different evaluation methods is as follows,

$$\hat{\mathcal{R}} := [M_3 \succ M_1 \succ M_2 \succ ... \succ M_m]. \tag{2}$$

The goal is to build an LLM ranking $\hat{\mathcal{R}}$ that aligns with human ranking $\mathcal{R}^*$, making the loss $\mathcal{L}$ of the both rankings tend towards 0, *i.e.*, $\mathcal{L}(\hat{\mathcal{R}}, \mathcal{R}^*) \to 0$

**Preference Alignment Metrics.** Before building LLM rankings, we first need to discuss how to evaluate aligned human rankings. Intuitively, the metrics we want mainly describe the differences between two arrays composed of ranking indices. Assuming that human ranking $\mathcal{R}^*$ is defined as being well-ranked in ascending order ($[1, 2, 3, ..., m]$) as shown in Eq 1. Thus the metric is to quantify the randomness of the learned ranking array ($[3, 1, 2, ..., m]$) as shown in Eq 2. Based on this, we propose three metrics called PEN, CIN, and LIS, respectively.

PEN (**P**ermutation **En**tropy). Permutation entropy [8] is a concept used to quantify the complexity or randomness of time series data. It provides a measure of the irregularity or unpredictability of the order of values in a sequence. We thus utilize it to measure the gap with human rankings as follows,

$$\mathcal{L}_{PEN}(\hat{\mathcal{R}}, \mathcal{R}^*) := -\sum p(\pi) \log p(\pi), \tag{3}$$

where

$$p(\pi) = \frac{\#\{t | 0 \le t \le m - k, (M_{t+1}, ..., M_{t+k}) \in \pi\}}{m - k + 1}.$$

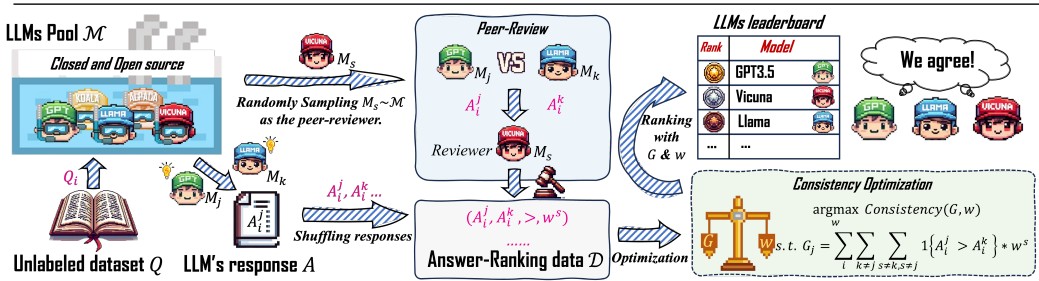

Figure 3: The pipeline of the PiCO. It is mainly composed of two components: the peer-review and consistency optimization stages. Specifically, in the peer-review stage, the unlabeled dataset $\mathcal{Q}$ and the LLMs pool $\mathcal{M}$ are given. Then, we let all LLMs answer each unlabeled question to obtain the response set $\mathcal{A}$. We shuffle the set and construct anonymous answer pairs, while randomly selecting other LLMs to evaluate both responses with a learnable confidence $w$. As a result, we can obtain the answer-ranking data $\mathcal{D}$ which is a quadruple that records the partial order between two answers and the evaluator's confidence weight. In the consistency optimization stage, we update the parameter $w$ by maximizing the consistency of each LLM's capability and score, while re-ranking the LLMs to be closer to human rankings.

$\pi$ denotes different permutations, $k$ is a hyper-parameter recommended to be set to 3 to 7, and we set $k = 3$ in this paper. Intuitively, it samples some subsequences and calculates the entropy for all permutation types. And the lower the permutation entropy in the learned LLM rankings, the closer it is to the ground-truth human rankings.

CIN (**C**ount **In**versions). Counting inversions [27] aims to measure the degree of disorder or "invertedness" in an array or sequence of elements. We thus define it as follows,

$$\mathcal{L}_{CIN}(\hat{\mathcal{R}}, \mathcal{R}^*) := \sum_{M_i, M_j \sim \mathcal{M}} \mathbf{1}\{M_i \succ M_j \wedge i < j\}. \tag{4}$$

Where $\mathbf{1}\{\cdot\}$ is the indicator function that the value is 1 when the condition is met, otherwise it is 0. Intuitively, the fewer inverse pairs in the learned LLM rankings, the closer it is to the ground-truth human rankings.

LIS (**L**ongest **I**ncreasing **S**ubsequence). The longest increasing subsequence aims to find the length of the longest subsequence in a given sequence of elements, where the subsequence is in increasing order. We utilize it to measure the degree of match with human rankings as follows,

$$\mathcal{L}_{LIS}(\hat{\mathcal{R}}, \mathcal{R}^*) := \max\{dp[i] \mid 1 \le i \le m\}, \tag{5}$$

where

$$dp[i] = 1 + \max\{dp[j] \mid 1 \le j < i \wedge M_j \prec M_i\}.$$

$dp[i]$ represents the length of the longest increasing subsequence that ends with $M_i$. LIS allows for a nuanced understanding of the degree to which the learned ranking aligns with the ideal human ranking, with a higher LIS length indicating greater alignment.

## 2.2 Algorithm Details

The PiCO framework, depicted in Figure 3, involves peer-review and consistency optimization stages. In the peer-review stage, we first collect an unlabeled dataset $\mathcal{Q}$ consisting of open-ended questions, and construct a large language model pool $\mathcal{M}$ that includes both open-source and closed-source LLMs. Then, we let all LLMs answer each unlabeled question to obtain the response set $\mathcal{A}$. We shuffle the set and construct anonymous answer pairs, while randomly selecting other LLMs as "reviewers" to evaluate both responses with a learnable confidence $w$. Finally, we can obtain the answer-ranking data $\mathcal{D}$ and calculate the response score $G$ for each large language model. In the consistency optimization phase, we maximize the consistency of each LLM's capability $w$ and score $G$ with constrained optimization, while re-ranking the LLMs to be closer to human rankings.

### 2.2.1 Peer Review Stage

**Data Collection and LLMs Pool Construction.** Benefiting from the creation of crowdsourced battle platforms, we accessed open assessment datasets from Chatbot Arena[55], MT-Bench[55],

and AlpacaEval[29]. These open datasets include critical fields such as "question_id" and "question_content." Utilizing the Chatbot Arena dataset, which features pairwise data from twenty LLMs with human preference annotations, we assembled an LLM pool $\mathcal{M} = \{M_j\}_{j=1}^m$. Leveraging 33K human-annotated interactions from this dataset, we established a ground-truth ranking $\mathcal{R}^*$ and gathered responses $\mathcal{A} = \{\{A_i^j\}_{i=1}^n\}_{j=1}^m$ for our dataset $\mathcal{Q} = \{Q_i\}_{i=1}^n$.

**Answer-Ranking Data Construction Based on Peer Review.** After obtaining the responses set $\mathcal{A}$, we aim to generate answer-ranking data $\mathcal{D}$ through the peer-review mechanism. Specifically, for the same question $Q_i \in \mathcal{Q}$, we randomly construct a battle pair $< A_i^j, A_i^k >$ for review. Each battle pair will be randomly assigned five models ("reviewers") to determine the winners or declare ties. Note that the model may evaluate its own answers, but the entire process is anonymous. As a result, we can obtain the quadruples $(A_i^j, A_i^k, > w^s)$, indicating the "reviewer" $M_s$ believes that the answer $A_i^j$ is better than answer $A_i^k$ with a confidence $w^s$. Therefore, the answer-ranking data $\mathcal{D}$ can be defined as follows,

$$\mathcal{D} = \left\{ (A_i^j, A_i^k, >, w^s) \right\}_{i \sim \mathcal{Q}, j,k,s \sim \mathcal{M}}, \tag{6}$$

where $i$ denotes the question index, and $j, k, s$ indicate the model indices. $w^s$ is a learnable confidence of model $M_s$, and $>$ is a partial order relationship from $\{>, <, =\}$.

### 2.2.2 Consistency Optimization Stage

As shown in Eq 6, following the peer-review mechanism, we construct anonymous answer pairs and randomly select other LLMs as "reviewers" to evaluate both responses with a learnable confidence $w$. Next, we expect to optimize the confidence $w$ and re-rank the LLMs to be closer to human rankings. We thus propose the consistency assumption, *i.e.*, high-level LLM can evaluate others' answers more accurately (confidence) than low-level ones, while higher-level LLM can also achieve higher answer-ranking scores. Formally, we maximize the consistency of each LLM's capability $w$ and score $G$ with constrained optimization as follows,

$$\underset{w}{\arg\max} \ \text{Consistency}(G, w) \tag{7}$$

$$\text{s.t. } G_j = \sum_{(A_i^j, A_i^k, >, w^s) \sim \mathcal{D}} \mathbf{1}\{A_i^j > A_i^k\} * w^s,$$

where $\mathbf{1}\{\cdot\}$ is the indicator function that the value is 1 when the condition is met, otherwise, it is 0. $G_j$ denotes the response score of model $M_j$, which is calculated by joint evaluation of other models. Moreover, we employ Pearson correlation [38] to measure the consistency between $w$ and $G$. Note that we only introduce this straightforward implementation to validate our idea of PiCO. Other more advanced strategies may be employed to further improve the performance.

**Discussion:** It is worth noting that the whole process (Eq. 6 and 7) works in an unsupervised way. The only thing we do is to adaptively assign each LLM a score that matches its abilities. An intuitive example is as follows: in a real peer-review system, if the academic level of three scholars $a$, $b$, and $c$ satisfies the following relationship, $w^a > w^b > w^c$. So, in the ultimate ideal scenario, the ranking of the scores submitted by these three scholars should also be, $G_a > G_b > G_c$. In other words, the sorting of $G$ and $w$ satisfies high consistency. On the other hand, scholars with stronger abilities (*i.e.*, scholar $a$) evaluate $A^b > A^c$ have stronger persuasiveness, so scholar $b$ should also receive higher weighted scores $1 * w^a$.

**Reviewer Elimination Mechanism.** Realizing that not all LLMs have sufficient ability to evaluate the responses of other models. We thus introduce an unsupervised elimination mechanism to remove those LLMs that have low scores. It iteratively removes the lowest-scoring LLM from the "reviewer queue" for the next consistency optimization stage, until 60% of models are eliminated. The whole process of the approach is summarized in Algorithm 1, and the details can be found in Appendix D.

## 3 Experiments

**Datasets.** To validate the effectiveness of the proposed approach, we perform experiments on Chatbot Arena[55], MT-Bench[55], and AlpacaEval[29]. The MT-Bench dataset assesses six LLMs' responses to 80 multi-category questions. The Chatbot Arena Conversations Dataset, with 33K conversations from 13K IPs during April-June 2023, evaluates real dialogue performance. AlpacaEval dataset

Table 1: Comparison of all methods on three datasets under data volumes of 1, 0.7 and 0.4, where the top value is highlighted by blod font. Lower PEN and CIN scores indicate better performance, while a higher LIS score signifies improved performance.

| Datasets | Chatbot Arena | | | MT-Bench | | | AlpacaEval | | |
|---|---|---|---|---|---|---|---|---|---|
| Methods | 1 | 0.7 | 0.4 | 1 | 0.7 | 0.4 | 1 | 0.7 | 0.4 |
| | PEN (↓) | | | | | | | | |
| Majority Voting [40] | $1.27^{\pm0.05}$ | $1.30^{\pm0.03}$ | $1.36^{\pm0.06}$ | $1.37^{\pm0.03}$ | $1.30^{\pm0.06}$ | $1.27^{\pm0.04}$ | $1.26^{\pm0.02}$ | $1.28^{\pm0.03}$ | $1.29^{\pm0.03}$ |
| Rating Voting [5] | $1.39^{\pm0.02}$ | $1.43^{\pm0.03}$ | $1.42^{\pm0.07}$ | $1.32^{\pm0.03}$ | $1.35^{\pm0.04}$ | $1.38^{\pm0.04}$ | $1.34^{\pm0.03}$ | $1.37^{\pm0.03}$ | $1.34^{\pm0.08}$ |
| GPTScore(flan-t5-xxl)[23] | $1.68^{\pm0.01}$ | $1.68^{\pm0.02}$ | $1.65^{\pm0.02}$ | $1.72^{\pm0.02}$ | $1.70^{\pm0.02}$ | $1.68^{\pm0.03}$ | $1.55^{\pm0.02}$ | $1.57^{\pm0.03}$ | $1.60^{\pm0.01}$ |
| GPTScore(davinci-002)[23] | $1.54^{\pm0.02}$ | $1.64^{\pm0.02}$ | $1.68^{\pm0.05}$ | $1.51^{\pm0.02}$ | $1.61^{\pm0.01}$ | $1.61^{\pm0.04}$ | $1.25^{\pm0.02}$ | $1.23^{\pm0.08}$ | $1.26^{\pm0.14}$ |
| PandaLM[46] | $1.65^{\pm0.01}$ | $1.64^{\pm0.02}$ | $1.63^{\pm0.05}$ | $1.55^{\pm0.03}$ | $1.59^{\pm0.05}$ | $1.52^{\pm0.08}$ | $1.56^{\pm0.01}$ | $1.58^{\pm0.01}$ | $1.64^{\pm0.05}$ |
| PRD[28] | $1.15^{\pm0.04}$ | $1.12^{\pm0.05}$ | $1.13^{\pm0.06}$ | $1.15^{\pm0.05}$ | $1.17^{\pm0.06}$ | $1.23^{\pm0.04}$ | $1.21^{\pm0.04}$ | $1.22^{\pm0.06}$ | $1.23^{\pm0.07}$ |
| PRE[17] | $1.07^{\pm0.01}$ | $1.03^{\pm0.03}$ | $1.06^{\pm0.04}$ | $1.17^{\pm0.04}$ | $1.13^{\pm0.05}$ | $1.19^{\pm0.05}$ | $1.18^{\pm0.03}$ | $1.21^{\pm0.04}$ | $1.15^{\pm0.05}$ |
| **PiCO (Ours)** | $\mathbf{0.94^{\pm0.02}}$ | $\mathbf{0.96^{\pm0.04}}$ | $\mathbf{0.95^{\pm0.08}}$ | $\mathbf{1.01^{\pm0.07}}$ | $\mathbf{1.02^{\pm0.11}}$ | $\mathbf{1.06^{\pm0.24}}$ | $\mathbf{1.17^{\pm0.02}}$ | $\mathbf{1.17^{\pm0.08}}$ | $\mathbf{1.13^{\pm0.05}}$ |
| | CIN (↓) | | | | | | | | |
| Majority Voting [40] | $22.00^{\pm0.00}$ | $23.25^{\pm1.09}$ | $25.00^{\pm2.55}$ | $23.00^{\pm0.00}$ | $20.50^{\pm0.87}$ | $21.00^{\pm1.00}$ | $20.00^{\pm0.00}$ | $21.25^{\pm1.30}$ | $22.25^{\pm1.30}$ |
| Rating Voting [5] | $24.00^{\pm0.00}$ | $24.50^{\pm1.29}$ | $25.00^{\pm1.15}$ | $22.00^{\pm0.00}$ | $22.50^{\pm1.00}$ | $24.25^{\pm0.50}$ | $22.00^{\pm0.00}$ | $22.50^{\pm0.58}$ | $22.50^{\pm1.00}$ |
| GPTScore(flan-t5-xxl)[23] | $67.00^{\pm0.00}$ | $66.50^{\pm0.50}$ | $68.25^{\pm1.09}$ | $53.00^{\pm0.00}$ | $55.75^{\pm2.77}$ | $54.50^{\pm2.29}$ | $35.00^{\pm0.00}$ | $36.00^{\pm0.71}$ | $37.75^{\pm1.60}$ |
| GPTScore(davinci-002)[23] | $42.00^{\pm0.00}$ | $45.50^{\pm1.12}$ | $51.00^{\pm5.61}$ | $33.00^{\pm0.00}$ | $35.00^{\pm0.71}$ | $36.25^{\pm1.64}$ | $21.00^{\pm0.00}$ | $20.25^{\pm2.86}$ | $21.50^{\pm4.39}$ |
| PandaLM[46] | $37.00^{\pm0.00}$ | $36.25^{\pm1.79}$ | $36.00^{\pm3.74}$ | $32.00^{\pm0.00}$ | $33.00^{\pm3.32}$ | $31.50^{\pm6.34}$ | $31.00^{\pm0.00}$ | $32.25^{\pm1.30}$ | $35.50^{\pm2.60}$ |
| PRD[28] | $17.00^{\pm0.00}$ | $16.25^{\pm0.43}$ | $17.50^{\pm1.50}$ | $17.00^{\pm0.00}$ | $17.75^{\pm1.09}$ | $19.50^{\pm1.50}$ | $19.00^{\pm0.00}$ | $19.25^{\pm1.48}$ | $19.50^{\pm0.87}$ |
| PRE[17] | $15.00^{\pm0.00}$ | $14.25^{\pm0.83}$ | $14.75^{\pm1.09}$ | $17.00^{\pm0.00}$ | $17.00^{\pm1.00}$ | $18.25^{\pm1.30}$ | $19.00^{\pm0.00}$ | $19.25^{\pm1.09}$ | $17.75^{\pm1.30}$ |
| **PiCO (Ours)** | $\mathbf{12.00^{\pm0.00}}$ | $\mathbf{12.50^{\pm0.50}}$ | $\mathbf{12.25^{\pm1.09}}$ | $\mathbf{14.50^{\pm0.50}}$ | $\mathbf{14.75^{\pm1.64}}$ | $\mathbf{16.00^{\pm6.36}}$ | $\mathbf{17.00^{\pm0.00}}$ | $\mathbf{18.00^{\pm1.87}}$ | $\mathbf{17.25^{\pm1.09}}$ |
| | LIS (↑) | | | | | | | | |
| Majority Voting [40] | $7.00^{\pm0.00}$ | $6.75^{\pm0.43}$ | $6.75^{\pm0.43}$ | $7.00^{\pm0.00}$ | $8.25^{\pm0.43}$ | $8.50^{\pm1.12}$ | $8.00^{\pm0.00}$ | $7.50^{\pm0.50}$ | $7.50^{\pm0.50}$ |
| Rating Voting [5] | $7.00^{\pm0.00}$ | $7.50^{\pm0.58}$ | $7.75^{\pm0.50}$ | $7.00^{\pm0.00}$ | $7.25^{\pm0.50}$ | $7.25^{\pm0.50}$ | $8.00^{\pm0.00}$ | $8.00^{\pm0.00}$ | $8.00^{\pm0.00}$ |
| GPTScore(flan-t5-xxl)[23] | $5.00^{\pm0.00}$ | $5.00^{\pm0.00}$ | $4.00^{\pm0.71}$ | $4.00^{\pm0.00}$ | $4.50^{\pm0.50}$ | $4.75^{\pm0.43}$ | $6.00^{\pm0.00}$ | $6.00^{\pm0.00}$ | $6.00^{\pm0.00}$ |
| GPTScore(davinci-002)[23] | $8.00^{\pm0.00}$ | $6.25^{\pm0.43}$ | $6.00^{\pm0.71}$ | $6.00^{\pm0.00}$ | $6.50^{\pm0.50}$ | $6.25^{\pm0.43}$ | $8.00^{\pm0.00}$ | $8.25^{\pm0.83}$ | $8.25^{\pm1.48}$ |
| PandaLM[46] | $5.00^{\pm0.00}$ | $5.50^{\pm0.50}$ | $6.00^{\pm0.00}$ | $6.00^{\pm0.00}$ | $7.00^{\pm0.71}$ | $7.25^{\pm0.43}$ | $6.00^{\pm0.00}$ | $5.75^{\pm0.43}$ | $5.50^{\pm0.50}$ |
| PRD[28] | $8.00^{\pm0.00}$ | $8.75^{\pm0.43}$ | $9.25^{\pm0.83}$ | $8.00^{\pm0.00}$ | $8.25^{\pm0.43}$ | $7.75^{\pm0.83}$ | $8.50^{\pm0.00}$ | $8.25^{\pm0.83}$ | $8.25^{\pm0.43}$ |
| PRE[17] | $9.00^{\pm0.00}$ | $10.25^{\pm0.43}$ | $10.00^{\pm0.87}$ | $8.00^{\pm0.00}$ | $8.50^{\pm0.50}$ | $8.25^{\pm0.83}$ | $8.00^{\pm0.00}$ | $8.00^{\pm0.00}$ | $8.25^{\pm0.43}$ |
| **PiCO (Ours)** | $\mathbf{10.00^{\pm0.00}}$ | $\mathbf{10.25^{\pm0.71}}$ | $\mathbf{10.50^{\pm0.43}}$ | $\mathbf{8.75^{\pm0.43}}$ | $\mathbf{8.75^{\pm0.87}}$ | $\mathbf{9.00^{\pm1.22}}$ | $\mathbf{9.00^{\pm0.00}}$ | $\mathbf{8.75^{\pm0.43}}$ | $\mathbf{8.50^{\pm0.50}}$ |

integrates 805 evaluations from diverse tests (e.g., Self-Instruct[48], OASST, Anthropic's helpful[7], Vicuna[16] and Koala[25] test sets) to align evaluations real-world interactions[21]. These datasets are collected by crowdsourcing platforms from human feedback, so they have a ground-truth ranking LLMs $\mathcal{R}^*$ aligned with human preferences.

**LLMs Pool.** In our experiments, we employ 15 LLMs with diverse architectures to construct the LLMs pool, including GPT-3.5-Turbo[35], WizardLM-13B[51], Guanaco-33B[1], Vicuna-7B[16], Vicuna-13B[16], Koala-13B[24], Mpt-7B[42], gpt4all-13B[6], ChatGLM-6B[53], Oasst-sft-4-pythia-12B[19], FastChat-T5-3B[55], StableLM-7B[3], Dolly-12B[18], LLaMA-13B[43], Alpaca-13B[41]. All models use the same evaluation template, they can be found in Appendix B

**Baselines.** To validate the effectiveness of the proposed PiCO approach, we compare the following methods in the experiments.

- *The wisdom of the crowds*: The two methods that perform LLMs evaluation based on the wisdom of the crowds [40, 13, 50] are compared in this experiment. *1)* **Majority Voting** [40]: Multiple review models vote for the better answer for the same response pair, and the model with the most votes gets 1 score; *2)* **Rating Voting** [5]: Multiple review models also vote on the same response pair, and the number of votes obtained is the score.

- *State-of-the-art methods*: The four recent SOTA methods of using either single or multiple models for self-evaluation are compared in this experiment. **PandaLM[46]:** It is a fine-tuned language model based on Llama-7b designed for the preference judgment tasks to evaluate and optimize LLMs. **GPTScore[23]:** It employs generative pre-trained models to assess the quality of generated text. It calculates the likelihood that the text was generated in response to specific instructions and context, indicative of high quality. In our implementation, GPT-3 (davinci-002) and flan-t5-xxl serve as the base models. **PRD[28]:** It transforms the LLMs win rates into weights for competitive ranking, while evaluating each LLM based on its preference for all possible pairs of answers, enabling a tournament-style ranking system. **PRE[17]:** It employs a supervised process to evaluate LLMs using a qualification exam, aggregates their scores based on accuracy, and assigns weights accordingly. **PiCO (Ours)**: the proposed approach in this paper.

**Metrics.** For all experiments, we employ three metrics to evaluate the aforementioned experimental setups and our Peer Review method: PEN, CIN, and LIS. Moreover, we perform the experiments for 4 runs and record the average results over 4 seeds ($seed = 1, 2, 3, 4$).

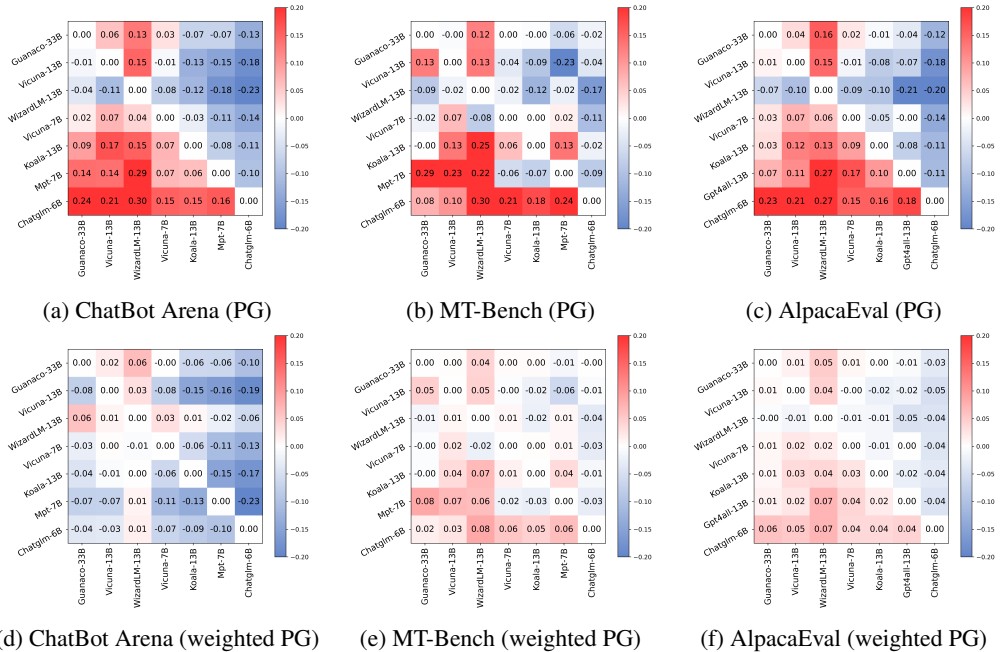

Figure 4: Heatmap distribution of preference gap (PG) metric among seven LLMs across three datasets. Higher values (above 0) indicate greater evaluation bias[17]. The first row shows original PG values in three datasets, while the second row displays PG values re-weighted using our learned confidence weights.

## 3.1 Performance Comparison

We validate the effectiveness of the proposed PiCO method on three datasets by comparing the following two types of methods, *i.e.*, the wisdom of the crowds and recent SOTA LLMs evaluation methods. The average results of PEN, CIN and LIS are demonstrated in Table 1. The ratios of response sets $\mathcal{D}$ are 1, 0.7, and 0.4, respectively.

The results presented in Table 1 illustrate the proposed PiCO method consistently surpasses competing approaches across the majority of evaluated metrics Notably, PiCO achieves performance improvements of 0.1, 2.5, and 0.92 on the PEN, CIN, and LIS metrics, respectively, compared to the Runner-up. These results underscore the superiority of aggregating evaluations from multiple models, such as Majority Voting, Rating Voting, PRD, and PRE, as opposed to relying solely on single-model methods like GPTScore and PandaLM. This collective model approach, leveraging 'the wisdom of the crowds', more accurately aligns with human rankings in our open-question evaluation framework.

In comparison with existing peer review evaluation methods(*i.e.,* PRD and PRE), it is evident that PiCO exhibits improvements across various evaluation metrics. Despite PRD's adjustment of model weights based on their win rates and PRE's reliance on supervised human feedback data to assign weights through a qualification exam, neither method achieves performance superior to the fully unsupervised PiCO approach. These methods rely on predefined criteria and human feedback, potentially leading to biases or suboptimal performance. In contrast, PiCO leverages unsupervised learning techniques, allowing it to autonomously adapt and discover patterns in the data without explicit human intervention.

It is important to highlight that PandaLM, a language model equipped with 7 billion parameters, was fine-tuned using labels generated by GPT-3.5-turbo as the ground truth, achieving stable performance across various datasets. However, in our unsupervised, open-ended experimental setup, which focuses on ranking-based metrics, GPTScore exhibits less robustness regardless of whether the base model is GPT-3 (davinci-002) or flan-t5-xx.

## 3.2 Exploring the Role of Confidence Weight

In this subsection, we will show that the confidence weight $w$ learned by our *consistency optimization* can reduce the system evaluation bias. Specifically, we first study whether the "review" model would

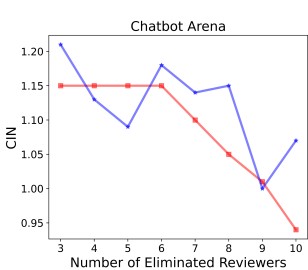
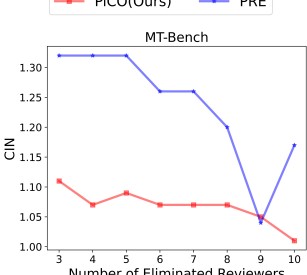
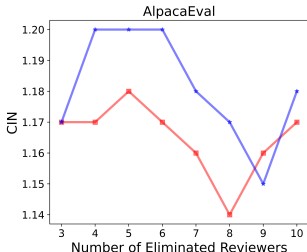

Figure 5: Performance comparison of the PiCO (Ours) and PRE[17] methods on the Chatbot Arena, MT-Bench, and AlpacaEval datasets, with the number of eliminated reviewers on the x-axis. The y-axis is CIN, where lower values indicate better performance.

prefer a particular model's response. Following [17], we employ the preference gap (PG) to evaluate the bias as follows,

$$PG(i, j) = P_i(i > j) - P_j(i > j), \tag{8}$$

where $P_i(i > j)$ represents the winning rate of model $i$ as the "reviewer" believes that $i$ defeated $j$. The heatmap distribution of the PG value $PG(i, j)$ among seven LLMs across three datasets is demonstrated in the first row of Figure 4. It can be observed that the evaluation system exhibits severe bias. Especially on ChatGLM-6B and Mpt-7B models, they often believe that their results are better than other ones, as their PG values are greater than 0 across three datasets.

After the *consistency optimization*, we assign the learned confidence weight $w$ to the corresponding model and ultimately obtain the re-weighting PG value $\hat{PG}(i, j)$ as follows,

$$\hat{PG}(i, j) = w_i \times P_i(i > j) - w_j \times P_j(i > j). \tag{9}$$

The results of the re-weighting PG value $\hat{PG}(i, j)$ are displayed on the second row of Figure 4. It can be observed that the learned confidence weight $w$ can significantly mitigate the preference gaps of the whole evaluation system. In our consistency optimization, LLMs such as ChatGLM-6B and Mpt-7B have lower weights, and reducing their confidence can effectively alleviate the system evaluation bias.

### 3.3 Study of Elimination Mechanism

The PiCO and PRE[17] methods both employ elimination mechanisms to remove those weakest LLMs from the "reviewer queue" during the evaluation process. As shown in Figure 5, the x-axis quantifies the number of reviewers eliminated, and the y-axis measures the CIN, where lower scores denote higher performance. Due to space limitations, more results on PEN and LIS metrics can be found in Appendix E. It can be observed that both PiCO and PRE exhibit better performance with an increasing number of eliminated "reviewers". The proposed PiCO approach can achieve better performance than PRE in most cases. It is worth noting that the PRE method employs the accuracy of "qualification exams" to eliminate weak LLMs, and this process requires human annotation [17]. On the contrary, the elimination process of our PiCO method is unsupervised and can still achieve better evaluation results than PRE.

### 3.4 Validation of Consistency Assumption

In this subsection, we conduct the ablation study to validate the effectiveness of the *consistency assumption*. Specifically, we first manually construct three methods: Forward Weight Voting, Uniform Weight Voting, and Reverse Weight Voting. That is, the ability weights of the model are respectively weighted forward ($w = [1, 0.9, ..., 0]$), uniformly ($w = [1, 1, ..., 1]$), and backward ($w = [0, 0.1, ..., 1]$) according to the ground-truth human ranking. Then, we randomly initialize the ability weights and employ our *consistency optimization* to adjust the weight. In addition, we also collect the average performance of "reviewer queue", *i.e.*, employing a single LLM as the "reviewer" to evaluate all response pairs and then calculate the average results of all LLMs.

As shown in Table 2, it can be observed that the Forward Weight Voting achieves better results than the Uniform and Backward ones in all cases, while the Backward one achieves worse results. It validates that assigning larger weights to those models with stronger capabilities can obtain better

Table 2: Ablation study comparing Backward, Uniform, Forward weight voting, and Consistency Optimization methods with the Average Performance of Reviewer Queue across three datasets.

| Methods | MT-Bench | | Chatbot Arena | | AlpacaEval | |
|---|---|---|---|---|---|---|
| | PEN ($\downarrow$) | CIN($\downarrow$) | PEN ($\downarrow$) | CIN($\downarrow$) | PEN ($\downarrow$) | CIN($\downarrow$) |
| Average Performance of Reviewer Queue | $1.49^{\pm0.28}$ | $34.87^{\pm14.68}$ | $1.49^{\pm0.26}$ | $38.80^{\pm19.28}$ | $1.50^{\pm0.23}$ | $33.13^{\pm13.97}$ |
| Backward Weight Voting | $1.43^{\pm0.04}$ | $25.00^{\pm0.00}$ | $1.43^{\pm0.05}$ | $26.00^{\pm0.00}$ | $1.36^{\pm0.03}$ | $24.00^{\pm0.00}$ |
| Uniform Weight Voting | $1.34^{\pm0.23}$ | $22.00^{\pm0.00}$ | $1.39^{\pm0.02}$ | $24.00^{\pm0.00}$ | $1.34^{\pm0.03}$ | $22.00^{\pm0.00}$ |
| Forward Weight Voting | $1.32^{\pm0.03}$ | $21.00^{\pm0.00}$ | $1.33^{\pm0.03}$ | $23.00^{\pm0.00}$ | $1.30^{\pm0.05}$ | $21.00^{\pm0.00}$ |
| Random Weight + Consistency Optimization | $\mathbf{1.17^{\pm0.06}}$ | $\mathbf{17.50^{\pm0.50}}$ | $\mathbf{1.20^{\pm0.08}}$ | $\mathbf{18.00^{\pm1.22}}$ | $\mathbf{1.21^{\pm0.04}}$ | $\mathbf{19.00^{\pm0.00}}$ |

results. Most importantly, employing our consistency optimization algorithm to assign weights to different review models can further improve the performance of the evaluation system, *i.e.*, lower PEN and CIN, as well as higher LIS in all cases. Moreover, it is worth noting that the average performance of the "reviewer queue" is very poor, even worse than the Backward Weight Voting. This means that the answer-ranking data $\mathcal{D}$ contains a lot of evaluation noise, while the proposed approach can still optimize weights and obtain better ranking results. In summary, the above experimental results validate the effectiveness of the consistency assumption from various perspectives.

# 4 Related Work

**Evaluation Benchmarks for Diversity.** LLMs are designed to handle a variety of tasks, necessitating comprehensive benchmarks[15]. Notable benchmarks include GLUE[45] and SuperGLUE[44], which simulate real-world scenarios across tasks such as text classification, translation, reading comprehension, and dialogue generation. HELM[30] provides a holistic evaluation of LLMs, assessing language understanding, generation, coherence, and reasoning. BIG-bench[39] pushes LLM capabilities with 204 diverse tasks. MMLU[26] measures multitask accuracy across domains like mathematics and law. However, these evaluations can be compromised by benchmark leakage, where evaluation data inadvertently used for training leads to inflated performance metrics[4, 56].

**Human Evaluation.** Human evaluation provides reliable feedback that closely aligns with real-world applications[15]. Liang et al.[30] evaluated summary and misinformation scenarios across multiple models. Ziems et al.[57] involved experts to assess model outputs in various domain-specific tasks. Bang et al.[9] examined ChatGPT's performance in summarization, translation, and reasoning using human-annotated datasets. The LMSYS initiative introduced platforms like Chatbot Arena[55], relying on human ratings as the primary evaluation metric. Despite its effectiveness, human evaluation is costly and subject to bias and cultural differences[37].

**Large Language Models for Evaluation.** The development of open-source LLMs has led to the use of LLMs as evaluators. GPTScore[23] uses models like GPT-3 to assign probabilities to high-quality content through multidimensional evaluation. Bubeck et al.[12] tested GPT-4, finding it rivaling human capabilities. Lin and Chen introduced LLM-EVAL[31] for evaluating dialogue quality with single prompts. PandaLM[46] employs LLMs as "judges" for evaluating instruction tuning. However, reliance on a single model can introduce biases such as positional[20], verbosity[47], and self-favoring biases[33, 55]. ChatEval[14] proposes a multi-agent framework to simulate human evaluation processes. Similarly, PRE[17] and PRD[28] use LLMs as evaluators, combining multiple evaluation outcomes for automated assessment. However, the PRE method, which relies on human feedback for supervised evaluation throughout the process, still incurs relatively high costs.

# 5 Conclusion

In this paper, we propose the novel Peer Review method based on the Consistency Optimization (PiCO) to automatically evaluate Large Language Models (LLMs) without relying on human feedback. PiCO utilizes *peer-review* mechanisms to autonomously assess LLMs in a shared environment, where both open-source and closed-source models can respond to unlabeled questions and evaluate each other. In this setup, each LLM's response score is determined collectively by other anonymous models, aiming to maximize consistency across capabilities and scores. We propose three metrics, *i.e.,* PEN, CIN, and LIS, to quantify the disparity from human preferences. The extensive experiment results across multiple datasets and metrics demonstrate that PiCO effectively generates an LLM leaderboard that aligns closely with human preferences. In the future, we plan to extend the peer-review mechanism to evaluate the capabilities of multi-modality large models.

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

# A Dataset Format

Focusing on the MT-Bench dataset, we demonstrate the ensuing data format utilizing dataset $\mathcal{Q}$. As Figure 6 illustrates, the Question dataset $\mathcal{Q}$ contains "Question id," "Category," "Question," and "Reference." In categories with definitive answers like "reasoning" or "math," the "Reference" field is populated with standard answers; otherwise, it remains blank. Each model M in our pool processes the Question dataset $\mathcal{Q}$ to generate the LLMs answer data $\mathcal{A}$, consisting of "Question id," "Answer id," "Model id," and "Answer." Finally, we combine pairs in $\mathcal{A}$ and appoint judges to evaluate, creating the Answer-Ranking data $\mathcal{D}$, featuring "Question id," "Model 1," "Model 2," "G1 winner," "G2 winner," and "Judge." Here, "G1 winner" and "G2 winner" indicate the outcomes of inputting reversed order responses of Model 1 and Model 2 into the judge model, a method employed to mitigate biases stemming from models' preferences for input order.

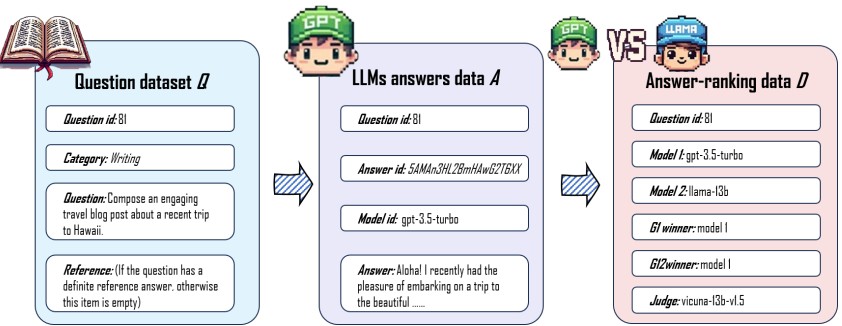

Figure 6: Format of the Question dataset $\mathcal{Q}$, LLMs responses data $\mathcal{A}$, and the Answer-Ranking data $\mathcal{D}$ for Peer Review

# B Detailed Prompt for Reviewers

The evaluation prompts, as detailed in Section 2.2.1, are employed during the Peer Review Stage. These prompts are provided to the Reviewer Language Model Systems (LLMs), enabling them to generate evaluative preferences. In our experimental framework, we devised four distinct prompt settings. For each setting, a tailored prompt template was meticulously crafted as illustrated below:

**Template for Single-Turn Interaction:** This template is designed for single-turn interactions between users and LLMs, where there is no predetermined correct answer. It facilitates open-ended dialogue, allowing for a wide range of user inquiries without the expectation of specific responses.

**Referenced Template for Single-Turn Interaction:** Tailored for single-turn dialogues between users and LLMs, this template incorporates predefined correct answers. It is particularly suited for

interactions involving factual inquiries, such as mathematics or logic problems, where accuracy and reference to correct information are paramount.

**Template for Multi-Turn Interaction:** This template caters to multi-turn conversations between users and LLMs, without predefined answers. It supports extended interactions, enabling users to explore topics in depth through a series of interconnected questions and responses.

**Referenced Template for Multi-Turn Interaction:** Designed for multi-turn dialogues with predefined correct answers, this template is ideal for complex inquiries requiring sequential reasoning or problem-solving, such as mathematical computations or logical deductions.

Each template is carefully constructed to match its intended use-case, providing a structured framework that guides the interaction between users and LLMs towards achieving desired outcomes, whether for open-ended exploration or precise problem-solving.

---

**Template for Single-Turn Answer**

**System prompt:** Please act as a judge and evaluate the quality of the responses provided by two AI assistants to the user question displayed below. You do not need to explain, just give your judgment. Output your final verdict by strictly following this format: "[[A]]" if assistant A is better, "[[B]]" if assistant B is better, and "[[C]]" for a tie.
**User Question:** {question}
**Assistant A's Answer:** {answer a}
**Assistant B's Answer:** {answer b}

---

**Referenced Template for Single-Turn Answer**

**System prompt:** Please act as a judge and evaluate the quality of the responses provided by two AI assistants to the user question displayed below, with reference to the provided reference answers. You do not need to explain, just give your judgment. Output your final verdict by strictly following this format: "[[A]]"if assistant A is better, "[[B]]" if assistant B is better, and "[[C]]" for a tie.
**User Question:** {question}
**Reference Answer:** {reference answer}
**Assistant A's Answer:** {answer a}
**Assistant B's Answer:** {answer b}

---

**Template for Multi-Turn Answer**

**System prompt:** Please act as a judge and evaluate the quality of the responses provided by two AI assistants to the user question displayed below. You do not need to explain, just give your judgment. Output your final verdict by strictly following this format: "[[A]]" if assistant A is better, "[[B]]" if assistant B is better, and "[[C]]" for a tie
**Assistant A's Conversation with User:**
    **User:** {question 1}
    **Assistant A:** {answer a1}
    **User:** {question 2}
    **Assistant A:** {answer a2}
**Assistant B's Conversation with User:**
    **User:** {question 1}
    **Assistant B:** {answer b1}
    **User:** {question 2}
    **Assistant B:** {answer b2}

## C   Scoring Methodology

In Section 2.2.2, Equation 7 delineates the methodology for optimizing scores. Within this framework, the function $\mathbf{1}\{A_i^j > A_i^k\}$ is more precisely defined as $f(A_i^j, A_i^k)$. Additionally, the function $f(A_i^j, A_i^k)$ is not fixed and can be implemented using various computational strategies. We introduce two distinct methodologies in this context: the Elo mechanism and the Rank mechanism.

Within the framework of the Elo mechanism, as specified by Equation 10, the $BASE$ value is set to 10, and the $SCALE$ factor is determined to be 400. This approach facilitates a dynamic adjustment of scores based on the outcomes of pairwise comparisons, allowing for a nuanced reflection of performance variations among models.

Conversely, in the context of the Rank mechanism, as outlined by Equation 11, $rank(j)$ signifies the current ranking of model $j$, with the constant $K$ assigned a value of 200. This mechanism employs a model's ranking within a predefined hierarchy as a pivotal factor in score calculation, thereby providing a straightforward, yet effective, method for evaluating comparative model performance.

$$f(A_i^j, A_i^k) = \begin{cases} 1 - \frac{1}{1+\text{BASE}^{((G(k)-G(j))/\text{SCALE})}} & \text{if } A_i^j > A_i^k \\ 0.5 - \frac{1}{1+\text{BASE}^{((G(k)-G(j))/\text{SCALE})}} & \text{if } A_i^j = A_i^k \\ 0 - \frac{1}{1+\text{BASE}^{((G(k)-G(j))/\text{SCALE})}} & \text{if } A_i^j < A_i^k \end{cases} \tag{10}$$

$$f(A_i^j, A_i^k) = \begin{cases} 1 + (rank(j) - rank(k))/K & \text{if } A_i^j > A_i^k \\ 0.5 & \text{if } A_i^j = A_i^k \\ 0 & \text{if } A_i^j < A_i^k \end{cases} \tag{11}$$

## D   Overall Algorithm of Peer Review

The overall algorithm, as delineated in Algorithm 1, encapsulates the comprehensive process outlined in Section 2.2. This sequence commences with "Data Collection and LLMs Pool Construction," progresses through "Answer-Ranking Data Construction Based on Peer Review," advances to "Consistency Optimization," and culminates with the "Unsupervised Elimination Mechanism."

**Algorithm 1** Overall Framework Algorithm of Peer Review

---

**Require:** Unlabeled dataset $\mathcal{Q}$, Pool of LLMs $\mathcal{M}$, Active LLM pool $\mathcal{M}^* = \mathcal{M}$
**Ensure:** Consistency-optimized ranking of LLMs $\mathcal{R}^*$
 1: Initialize response matrix $A \leftarrow \emptyset$
 2: **for** each question $q_i \in \mathcal{Q}$ **do**
 3:     Initialize response vector for question $q_i$, $A^i \leftarrow \emptyset$
 4:     **for** each model $m_j \in \mathcal{M}$ **do**
 5:         $A^i_j \leftarrow$ response of model $m_j$ to question $q_i$
 6:         $A^i \leftarrow A^i \cup \{A^i_j\}$
 7:     **end for**
 8:     Shuffle $A^i$ to obtain permuted response vector $A^i$
 9:     $A \leftarrow A \cup \{A^i\}$
10: **end for**
11: Initialize answer-ranking data $D \leftarrow \emptyset$
12: Initialize model weights vector $w$ with Gaussian distribution
13: **for** each permuted response vector $A^i$ **do**
14:     **for** each pair of responses $(A^j_i, A^k_i)$ in $A^i$ **do**
15:         **for** $s \leftarrow 1$ to 5 **do**                    ▷ Randomly select 5 models for evaluation
16:             Evaluate the pair $(A^j_i, A^k_i)$ with model $m_s$
17:             $D \leftarrow D \cup \{(A^j_i, A^k_i, > w^s)\}$
18:         **end for**
19:     **end for**
20: **end for**
21: Initialize scores $G_j$ for each model $m_j \in \mathcal{M}$ to the Elo initial score
22: **repeat**
23:     **while** not converged **do**
24:         **for** each model $m_j \in \mathcal{M}$ **do**
25:             Compute $G_j$ using updated formula:
26:             $G_j = \sum_i \sum_{k \neq j} \sum_{s \neq k, s \neq j} \mathbf{1}\{A^j_i, A^k_i\} \times w^s \quad (A^j_i, A^k_i, > w^s, s \in \mathcal{M}^*) \in D$
27:         **end for**
28:         Update weight vector $w$ to maximize the consistency of $w$ and $G$
29:     **end while**
30:     Sort $\mathcal{M}^*$ by $G_j$ to identify $\mathcal{M}_{min}$, the lowest-scoring model
31:     **if** size of $\mathcal{M}^* >$ threshold **then**
32:         Remove $\mathcal{M}_{min}$ from $\mathcal{M}^*$
33:     **end if**
34: **until** size of $\mathcal{M}^* <$ threshold
35: Compute the final ranking $\mathcal{R}^*$ based on the optimized scores $G_j$
36: **return** $\mathcal{R}^*$

---

# E   Complete Experimental Results

In Section 3.4, we both employ elimination mechanisms to cull the weakest LLMs from the 'reviewer queue' during the evaluation process. In Figures 7 and 8, we present the results for the PEN and LIS metrics, where lower PEN scores indicate better performance, and higher LIS scores denote superior performance. It is evident that both the 'PiCO' and PRE approaches demonstrate enhanced performance as the number of eliminated 'reviewers' increases. In most cases, the proposed 'PiCO' method outperforms PRE.

In Section 3.5, we validate the effectiveness of the *consistency assumption* and compare it with the Average Performance of the Reviewer Queue, i.e., employing a single LLM as the 'reviewer' to evaluate all response pairs and then calculating the average results of all LLMs. The comprehensive results compared with the Reviewer Queue are illustrated in Table3, Figure 9, 10 and 11, revealing that in the full Reviewer Queue, the performance of the vast majority of LLMs is very poor, indicating that the evaluations from most LLMs are noise. However, our 'PiCO' approach nearly matches the evaluative prowess of the pool's most capable LLM, GPT-3.5. Remarkably, given its unsupervised nature, the 'PiCO' method demonstrates the capability to mitigate the influence of noise, reaching the

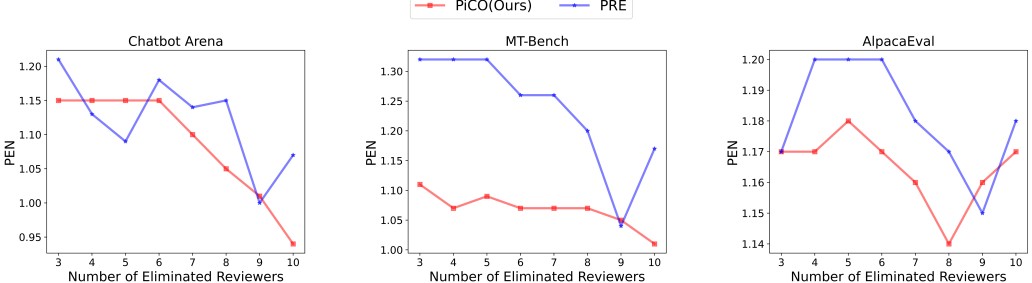

Figure 7: Performance comparison of the PiCO (Ours) and PRE[17] methods on the MT-Bench, Chatbot Arena, and AlpacaEval datasets, with the number of eliminated reviewers on the x-axis. The y-axis is PEN, where lower values indicate better performance.

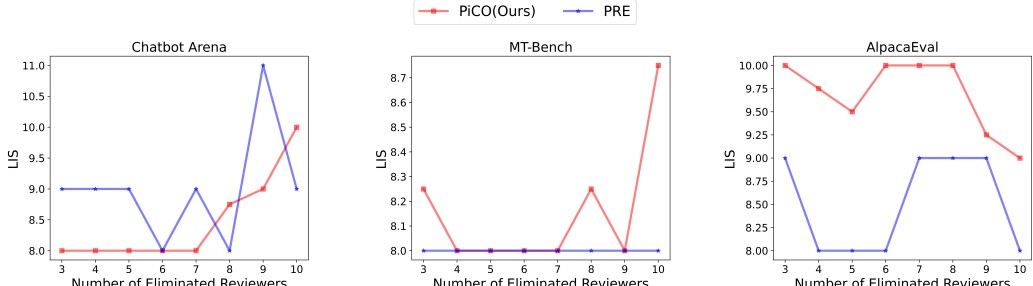

Figure 8: Performance comparison of the PiCO (Ours) and PRE[17] methods on the MT-Bench, Chatbot Arena, and AlpacaEval datasets, with the number of eliminated reviewers on the x-axis. The y-axis is LIS, where upper values indicate better performance.

Table 3: Comparison of performance across three datasets using Unsupervised methods versus using single models in reviewer queue.

| Methods | MT-Bench | | | Chatbot Arena | | | AlpacaEval | | |
|---|---|---|---|---|---|---|---|---|---|
| | PEN ($\downarrow$) | CIN($\downarrow$) | LIS($\uparrow$) | PEN ($\downarrow$) | CIN($\downarrow$) | LIS($\uparrow$) | PEN ($\downarrow$) | CIN($\downarrow$) | LIS($\uparrow$) |
| Gpt-3.5 | **0.97** | **12.00** | **10.00** | **0.85** | **11.00** | **11.00** | **1.15** | **16.00** | **9.00** |
| Guanaco-33B | 1.25 | 21.00 | 8.00 | 1.50 | 28.00 | 7.00 | 1.26 | 20.00 | 9.00 |
| Vicuna-13B | 1.31 | 20.00 | 7.00 | 1.27 | 23.00 | 8.00 | 1.20 | 17.00 | 8.00 |
| WizardLM-13B | 1.15 | 17.00 | 9.00 | 1.27 | 19.00 | 8.00 | 1.17 | 17.00 | 9.00 |
| Vicuna-7B | 1.27 | 21.00 | 8.00 | 1.30 | 20.00 | 7.00 | 1.34 | 23.00 | 8.00 |
| Koala-13B | 1.67 | 43.00 | 6.00 | 1.34 | 23.00 | 8.00 | 1.54 | 31.00 | 7.00 |
| gpt4all-13B | 1.74 | 45.00 | 6.00 | 1.60 | 35.00 | 6.00 | 1.73 | 42.00 | 6.00 |
| Mpt-7B | 1.67 | 39.00 | 6.00 | 1.72 | 52.00 | 6.00 | 1.63 | 34.00 | 7.00 |
| Oass-pythia-12B | 1.77 | 50.00 | 5.00 | 1.74 | 42.00 | 5.00 | 1.70 | 47.00 | 6.00 |
| Alpaca-13B | 1.77 | 49.00 | 7.00 | 1.60 | 73.00 | 4.00 | 1.63 | 34.00 | 7.00 |
| FastChat-T5-3B | 1.45 | 29.00 | 7.00 | 1.53 | 30.00 | 7.00 | 1.30 | 22.00 | 7.00 |
| ChatGLM-6B | 1.59 | 33.00 | 7.00 | 1.71 | 55.00 | 5.00 | 1.63 | 34.00 | 6.00 |
| StableLM-7B | 1.68 | 63.00 | 5.00 | 1.75 | 44.00 | 5.00 | 1.72 | 56.00 | 4.00 |
| Dolly-12B | 1.76 | 46.00 | 6.00 | 1.57 | 71.00 | 6.00 | 1.75 | 54.00 | 6.00 |
| LLaMA-13B | 1.60 | 35.00 | 7.00 | 1.76 | 56.00 | 6.00 | 1.70 | 50.00 | 5.00 |
| Average Performance of All Review LLMs | 1.51 | 34.87 | 6.93 | 1.50 | 38.80 | 6.60 | 1.50 | 33.13 | 6.93 |
| PRD[28] | 1.15 | 17.00 | 8.00 | 1.15 | 17.00 | 8.00 | 1.21 | 19.00 | 9.00 |
| PRE[17] | 1.17 | 17.00 | 8.00 | 1.07 | 15.00 | 9.00 | 1.18 | 19.00 | 8.00 |
| PiCO (Ours) | 1.01 | 14.50 | 8.75 | 0.94 | 12.00 | 10.00 | 1.17 | 17.00 | 9.00 |

evaluation upper bound (the strongest LLM) within any given unknown LLM pool $M$, even in the absence of prior ranking information.

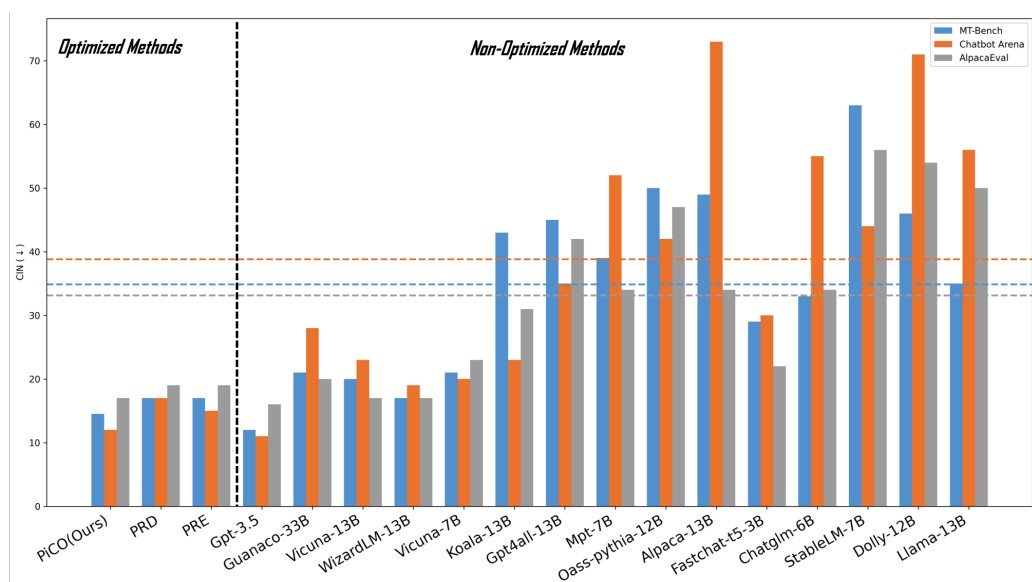

Figure 9: Comparison of performance on the CIN metric across three datasets using Unsupervised methods versus using single models, with Unsupervised methods on the left and Supervised methods on the right. The dotted line represents the average value using single models.

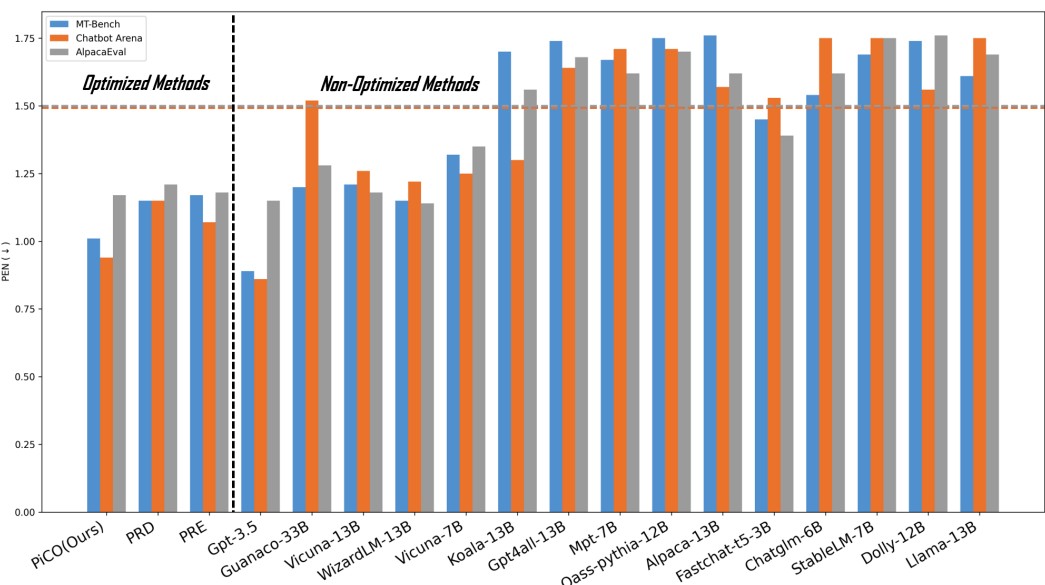

Figure 10: Comparison of performance on the PEN metric across three datasets using Unsupervised methods versus using single models, with Unsupervised methods on the left and Supervised methods on the right. The dotted line represents the average value using single models.

## F Selected Models and Optimized Ranking

For our analysis, we meticulously selected 15 LLMs spanning a variety of architectures, encompassing both open-source and closed-source models, as detailed in the subsequent table. Our curated selection features prominent LLMs including the closed-source "gpt-3.5-turbo," "chatglm" which is predicated on the encoder-decoder framework, "fastchat-t5-3b" that leverages Google's T5 (Text-to-Text Transfer Transformer) architecture, and "llama-13b" founded on the GPT architectural principles.

We have comprehensively detailed the ranking outcomes across three distinct datasets for our comparative analysis, incorporating the optimized model rankings, names, and their respective scores.

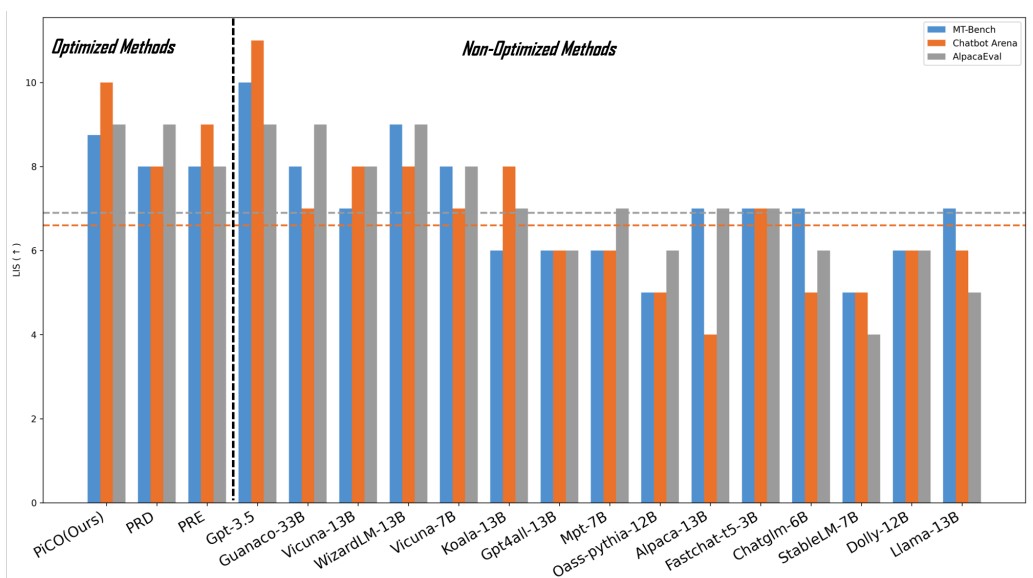

Figure 11: Comparison of performance on the LIS metric across three datasets using Unsupervised methods versus using single models, with Unsupervised methods on the left and Supervised methods on the right. The dotted line represents the average value using single models.

As delineated in Appendix C, the PiCO (Ours) is capable of employing various scoring mechanisms, thereby facilitating the presentation of ranking outcomes on three datasets utilizing both the Elo and Rank mechanisms. Furthermore, we have also enumerated the ranking results for PRD and PRE methodologies across the three datasets, offering a holistic view of the competitive landscape.

 **F.1  PiCO**

---

**Grade-Elo-Chatbot**

#1 **Gpt-3.5** | Grade: 9205.162109375
#2 **WizardLM-13B** | Grade: 9143.46875
#3 **Guanaco-33B** | Grade: 5886.92626953125
#4 **Vicuna-7B** | Grade: 5368.9462890625
#5 **Vicuna-13B** | Grade: 5216.79541015625
#6 **Koala-13B** | Grade: 3545.1171875 | Eliminated
#7 **Mpt-7B** | Grade: 962.99462890625 | Eliminated
#8 **Gpt4all-13B** | Grade: 652.4602661132812 | Eliminated
#9 **Chatglm-6B** | Grade: 417.1375427246094 | Eliminated
#10 **Oasst-pythia-12B** | Grade: -898.2676391601562 | Eliminated
#11 **Fastchat-t5-3B** | Grade: -1251.7183837890625 | Eliminated
#12 **StableLM-7B** | Grade: -2232.66943359375 | Eliminated
#13 **Dolly-12B** | Grade: -3163.540283203125 | Eliminated
#14 **Llama-13B** | Grade: -3648.37841796875 | Eliminated
#15 **Alpaca-13B** | Grade: -14204.3984375 | Eliminated

---

**Grade-Elo-AlpacaEval**

#1 **WizardLM-13B** | Grade: 8662.7158203125
#2 **Vicuna-13B** | Grade: 5586.46630859375
#3 **Guanaco-33B** | Grade: 5445.341796875
#4 **Vicuna-7B** | Grade: 5374.2314453125
#5 **Gpt-3.5** | Grade: 4845.91552734375
#6 **Koala-13B** | Grade: 4338.77783203125 | Eliminated
#7 **Chatglm-6B** | Grade: 2293.4208984375 | Eliminated
#8 **Gpt4all-13B** | Grade: 2080.511962890625 | Eliminated
#9 **Mpt-7B** | Grade: 1694.4945068359375 | Eliminated
#10 **Fastchat-t5-3B** | Grade: 1371.94287109375 | Eliminated
#11 **Oasst-pythia-12B** | Grade: -665.8685302734375 | Eliminated
#12 **StableLM-7B** | Grade: -1343.5838623046875 | Eliminated
#13 **Dolly-12B** | Grade: -5377.13427734375 | Eliminated
#14 **Llama-13B** | Grade: -5847.59130859375 | Eliminated
#15 **Alpaca-13B** | Grade: -13459.6162109375 | Eliminated

---

**Grade-Elo-MT_Bench**

#1 **WizardLM-13B** | Grade: 2178.10302734375
#2 **Vicuna-13B** | Grade: 1720.1114501953125
#3 **Guanaco-33B** | Grade: 1704.1832275390625
#4 **Vicuna-7B** | Grade: 1659.2799072265625
#5 **Gpt-3.5** | Grade: 1535.8819580078125
#6 **Mpt-7B** | Grade: 1338.5235595703125 | Eliminated
#7 **Koala-13B** | Grade: 1267.9747314453125 | Eliminated
#8 **Chatglm-6B** | Grade: 1011.7701416015625 | Eliminated
#9 **Gpt4all-13B** | Grade: 976.5963745117188 | Eliminated
#10 **Oasst-pythia-12B** | Grade: 779.3573608398438 | Eliminated
#11 **StableLM-7B** | Grade: 512.1678466796875 | Eliminated
#12 **Alpaca-13B** | Grade: 334.9879455566406 | Eliminated
#13 **Fastchat-t5-3B** | Grade: 303.5980529785156 | Eliminated
#14 **Dolly-12B** | Grade: 72.63818359375 | Eliminated
#15 **Llama-13B** | Grade: -395.19921875 | Eliminated

**Grade-Rank-Chatbot**

#1 **WizardLM-13B** | Grade: 0.30809280276298523
#2 **Gpt-3.5** | Grade: 0.293962299823761
#3 **Guanaco-33B** | Grade: 0.28587597608566284
#4 **Vicuna-7B** | Grade: 0.28212910890579224
#5 **Vicuna-13B** | Grade: 0.27900218963623047
#6 **Koala-13B** | Grade: 0.2672431766986847 | Eliminated
#7 **Mpt-7B** | Grade: 0.2500302195549011 | Eliminated
#8 **Gpt4all-13B** | Grade: 0.24746862053871155 | Eliminated
#9 **Chatglm-6B** | Grade: 0.2466953843832016 | Eliminated
#10 **Oasst-pythia-12B** | Grade: 0.23637069761753082 | Eliminated
#11 **Fastchat-t5-3B** | Grade: 0.2350562959909439 | Eliminated
#12 **StableLM-7B** | Grade: 0.22843806445598602 | Eliminated
#13 **Dolly-12B** | Grade: 0.22219440340995789 | Eliminated
#14 **Llama-13B** | Grade: 0.2165679931640625 | Eliminated
#15 **Alpaca-13B** | Grade: 0.13975904881954193 | Eliminated

562

**Grade-Rank-AlpacaEval**

#1 **WizardLM-13B** | Grade: 0.4019235074520111
#2 **Vicuna-13B** | Grade: 0.36745429039001465
#3 **Guanaco-33B** | Grade: 0.3664878010749817
#4 **Vicuna-7B** | Grade: 0.36541733145713806
#5 **Gpt-3.5** | Grade: 0.36000365018844604
#6 **Koala-13B** | Grade: 0.3544933795928955 | Eliminated
#7 **Chatglm-6B** | Grade: 0.3319571018218994 | Eliminated
#8 **Gpt4all-13B** | Grade: 0.3306528627872467 | Eliminated
#9 **Mpt-7B** | Grade: 0.32641729712486267 | Eliminated
#10 **Fastchat-t5-3B** | Grade: 0.32173293828964233 | Eliminated
#11 **Oasst-pythia-12B** | Grade: 0.2999681532382965 | Eliminated
#12 **StableLM-7B** | Grade: 0.2932431995868683 | Eliminated
#13 **Dolly-12B** | Grade: 0.24777530133724213 | Eliminated
#14 **Llama-13B** | Grade: 0.24381506443023682 | Eliminated
#15 **Alpaca-13B** | Grade: 0.16114839911460876

563

**Grade-Rank-MT_Bench**

#1 **WizardLM-13B** | Grade: 0.2994651198387146
#2 **Vicuna-13B** | Grade: 0.2809261679649353
#3 **Guanaco-33B** | Grade: 0.2767307460308075
#4 **Vicuna-7B** | Grade: 0.2758147716522217
#5 **Gpt-3.5** | Grade: 0.27261608839035034
#6 **Mpt-7B** | Grade: 0.26338690519332886 | Eliminated
#7 **Koala-13B** | Grade: 0.2613368630409241 | Eliminated
#8 **Gpt4all-13B** | Grade: 0.24908888339996338 | Eliminated
#9 **Chatglm-6B** | Grade: 0.24898234009742737 | Eliminated
#10 **Oasst-pythia-12B** | Grade: 0.2415400892496109 | Eliminated
#11 **StableLM-7B** | Grade: 0.2299075722694397 | Eliminated
#12 **Alpaca-13B** | Grade: 0.22171474993228912 | Eliminated
#13 **Fastchat-t5-3B** | Grade: 0.221677765250206 | Eliminated
#14 **Dolly-12B** | Grade: 0.21185410022735596 | Eliminated
#15 **Llama-13B** | Grade: 0.192665234208107 | Eliminated

564

## F.2    PRD

---

**PRD-Chatbot**

---

#1 **WizardLM-13B** | Grade: 5565.28271484375
#2 **Gpt-3.5** | Grade: 4613.22900390625
#3 **Guanaco-33B** | Grade: 3423.588134765625
#4 **Vicuna-7B** | Grade: 2985.4892578125
#5 **Vicuna-13B** | Grade: 2972.15673828125
#6 **Koala-13B** | Grade: 2237.70751953125
#7 **Chatglm-6B** | Grade: 875.373779296875
#8 **Mpt-7B** | Grade: 602.46923828125
#9 **Gpt4all-13B** | Grade: 356.06243896484375
#10 **Fastchat-t5-3B** | Grade: 184.89663696289062
#11 **Dolly-12B** | Grade: 52.10746765136719
#12 **Oasst-pythia-12B** | Grade: -307.49908447265625
#13 **StableLM-7B** | Grade: -691.4453735351562
#14 **Llama-13B** | Grade: -848.1654052734375
#15 **Alpaca-13B** | Grade: -7020.923828125

---

**PRD-AlpacaEval**

---

#1 **WizardLM-13B** | Grade: 5469.75634765625
#2 **Guanaco-33B** | Grade: 3707.014892578125
#3 **Vicuna-13B** | Grade: 3618.63427734375
#4 **Vicuna-7B** | Grade: 3569.389892578125
#5 **Gpt-3.5** | Grade: 3197.755615234375
#6 **Koala-13B** | Grade: 2893.642578125
#7 **Chatglm-6B** | Grade: 1847.1300048828125
#8 **Fastchat-t5-3B** | Grade: 1585.66943359375
#9 **Gpt4all-13B** | Grade: 1561.145751953125
#10 **Mpt-7B** | Grade: 1332.3753662109375
#11 **StableLM-7B** | Grade: -33.00855255126953
#12 **Oasst-pythia-12B** | Grade: -92.68387603759766
#13 **Dolly-12B** | Grade: -3013.588623046875
#14 **Llama-13B** | Grade: -3211.0302734375
#15 **Alpaca-13B** | Grade: -7432.3701171875

---

**PRD-MT_Bench**

---

#1 **WizardLM-13B** | Grade: 1811.64697265625
#2 **Vicuna-13B** | Grade: 1537.8084716796875
#3 **Guanaco-33B** | Grade: 1481.1739501953125
#4 **Vicuna-7B** | Grade: 1401.5194091796875
#5 **Gpt-3.5** | Grade: 1272.8072509765625
#6 **Mpt-7B** | Grade: 1186.5518798828125
#7 **Chatglm-6B** | Grade: 1166.6246337890625
#8 **Koala-13B** | Grade: 1124.2513427734375
#9 **Gpt4all-13B** | Grade: 871.2874755859375
#10 **Oasst-pythia-12B** | Grade: 855.3653564453125
#11 **StableLM-7B** | Grade: 782.702880859375
#12 **Fastchat-t5-3B** | Grade: 636.966064453125
#13 **Alpaca-13B** | Grade: 414.9374694824219
#14 **Dolly-12B** | Grade: 377.5018005371094
#15 **Llama-13B** | Grade: 78.90127563476562

## F.3 PRE

---

**PRE-Chatbot**

---

#1 **WizardLM-13B** | Grade: 1113.7034715479742
#2 **Gpt-3.5** | Grade: 1076.1116664199608
#3 **Guanaco-33B** | Grade: 1067.441581415147
#4 **Vicuna-13B** | Grade: 1057.702184441485
#5 **Vicuna-7B** | Grade: 1043.4840340151043
#6 **Koala-13B** | Grade: 1030.4455842017508 | Eliminated
#7 **Chatglm-6B** | Grade: 1012.4487557424748 | Eliminated
#8 **Mpt-7B** | Grade: 1000.487230109001 | Eliminated
#9 **Gpt4all-13B** | Grade: 1000.4111397038492 | Eliminated
#10 **Fastchat-t5-3B** | Grade: 992.3732179832363 | Eliminated
#11 **Oasst-pythia-12B** | Grade: 977.5217305871272 | Eliminated
#12 **StableLM-7B** | Grade: 970.3665926795535 | Eliminated
#13 **Llama-13B** | Grade: 929.6268868888149 | Eliminated
#14 **Dolly-12B** | Grade: 929.1943463130976 | Eliminated
#15 **Alpaca-13B** | Grade: 798.6815779514078 | Eliminated

---

**PRE-AlpacaEval**

---

#1 **WizardLM-13B** | Grade: 1127.822808841937
#2 **Vicuna-7B** | Grade: 1077.1823389450524
#3 **Vicuna-13B** | Grade: 1075.4338443616266
#4 **Guanaco-33B** | Grade: 1074.8043135229418
#5 **Gpt-3.5** | Grade: 1065.305736105376
#6 **Gpt4all-13B** | Grade: 1039.4091630861865 | Eliminated
#7 **Koala-13B** | Grade: 1038.205749976473 | Eliminated
#8 **Mpt-7B** | Grade: 1032.2893401162178 | Eliminated
#9 **Chatglm-6B** | Grade: 1027.1937496918501 | Eliminated
#10 **Fastchat-t5-3B** | Grade: 992.3481168791307 | Eliminated
#11 **StableLM-7B** | Grade: 979.3894141445692 | Eliminated
#12 **Oasst-pythia-12B** | Grade: 940.6438439723215 | Eliminated
#13 **Dolly-12B** | Grade: 886.1412110662756 | Eliminated
#14 **Llama-13B** | Grade: 880.0797724297793 | Eliminated
#15 **Alpaca-13B** | Grade: 763.7505968602533 | Eliminated

---

**PRE-MT_Bench**

---

#1 **WizardLM-13B** | Grade: 1065.5843776639435
#2 **Vicuna-13B** | Grade: 1062.3934138040302
#3 **Guanaco-33B** | Grade: 1052.2206466556906
#4 **Vicuna-7B** | Grade: 1035.1112817247572
#5 **Gpt-3.5** | Grade: 1029.8316754711038
#6 **Koala-13B** | Grade: 1024.9307662983267 | Eliminated
#7 **Chatglm-6B** | Grade: 1020.5238960907612 | Eliminated
#8 **Mpt-7B** | Grade: 1014.0683255081057 | Eliminated
#9 **Gpt4all-13B** | Grade: 991.7142639623017 | Eliminated
#10 **StableLM-7B** | Grade: 979.8443261256327 | Eliminated
#11 **Oasst-pythia-12B** | Grade: 977.9930430111322 | Eliminated
#12 **Fastchat-t5-3B** | Grade: 953.0776159143571 | Eliminated
#13 **Alpaca-13B** | Grade: 949.129770731626 | Eliminated
#14 **Dolly-12B** | Grade: 928.511065779112 | Eliminated
#15 **Llama-13B** | Grade: 915.0655312591185 | Eliminated

