# OpenReview forum: "PiCO: Peer Review in LLMs based on the Consistency Optimization"
_NeurIPS.cc/2024/Conference — Submitted to NeurIPS 2024_

### Official Review · Reviewer_e5pS · 2024-07-05

**Soundness:** 3
**Presentation:** 3
**Contribution:** 3
**Rating:** 6
**Confidence:** 3

**Summary:**

The paper introduces a novel unsupervised evaluation method for large language models (LLMs): it uses a peer-review mechanism of a models' anonymized answers by other models. The approach assigns a (learnable) capability parameter to each LLM and solves a constrained optimization problem to maximize the consistency between capabilities and scores. The end result is a ranking of the evaluated models.

**Strengths:**

The paper introduce a novel approach to an important practical problem: ranking the quality of the ever-growing number of open- and closed- source LLM available to the public. The paper is reasonably easy to follow, and the empirical results appear to be sound.

**Weaknesses:**

The paper could be further improved on several directions:
1) you should dedicate a full section to the iterative elimination of models; what is the benefit of eliminating the weaker ones rather than keeping them around? how di you come up with the threshold of 60% to remove? can you learn this threshold automatically? is this threshold optimal for these 15 models? what happens if you start with, say, 100 models? what happens to your results (and the curves in Fig 5) if you stop earlier (all three metrics, not just CIN)? What if you continue to eliminate all models until you are left with one? is there any relationship between the order in which the models are eliminated and their final rank?
2) are there any scaling issues for 100, 1K, 10K, or 100K models? how about cost: is it cheaper to fine-tune a "baseline" model than to pick the best one out of 10K candidates?
3) while the three metrics you use are meaningful, you should also present results for Precision@1 and -say- RBP@3; after all, we care a lot about identifying the the top models
3) Fig 5 should be extended to nine graphs (3 metrics * 3 datasets); for each of the 9 graphs, you should also show illustrative three ranked lists: PiCO's, PRE's, and the target one. As always, the devil is in the details: not all "CIN = 1" are created equal. Performance-wise, it is almost irrelevant if you have the bottom-2 models inverted; no necessarily so if the inversion is between the top-2 models

**Questions:**

1. line 144 - what is the benefit of a model evaluating itself? why not explicitly disallow it? w/o a full explanation, this detail unnecessarily  complicates the story
2. you repeatedly talk about "higher-level LLM" w/o defining the concept; is it based on the number of parameters (larger --> higher)?
3. can PiCO (or an extension) also estimate the gap between the performance of two models that are adjacent in the final ranking?

NITPICKS
- line 72: "closer" ... than whom? existing approaches, I presume
- line 154: the "quadruple" appears to be a "triple"; adding "s" would make it a 4-tuple
- you talk a few times about "persuasiveness;" this concept makes sense for human reviewers discussing a submitted article, but it is unclear why it matters for LLMs
- the related work section should be right after the Intro

---

> ### Author Rebuttal · Authors · 2024-08-07
>
> Thank you for your thoughtful comments, which prompt us to improve our manuscript.
>
> **Q1: What is the benefit of eliminating the weaker ones rather than keeping them around? how did you come up with the threshold of 60\% to remove? can you learn this threshold automatically?**
>
> **A**: The benefit of eliminating the weaker ones rather than keeping them around can make the peer-reviewer system more robust. We found that the weaker LLMs also have worse evaluation ability, which will introduce much evaluation noise to damage the peer-review system. We plot the average training loss curve with the increase of eliminated reviewers in **Figure R3 of the additional rebuttal material**. It can be observed that when eliminating weaker reviewers, the average loss of the whole peer-review system is reduced. It shows that removing those noise evaluations will benefit the whole system. On the other hand, when 60\% ($=9$) weaker reviewers are eliminated, the whole system loss achieves the lowest. **This phenomenon occurs across three datasets, which indicates that the eliminating threshold is automatically learned**. In addition, removing more strong reviewers ($>9$) will also hurt the evaluation process. We appreciate your valuable review, and we will add these results and discussions in our revised paper.
>
> **Q2: Are there any scaling issues for 100, 1K, 10K, or 100K models? how about the cost?**
>
> **A**: In our experiments, we validate the scaling issues of the proposed PiCO approach from another perspective. As shown in **Table 1 (main text) and R1 (global author rebuttal)**, we randomly used 100\%, 70\%, 40\%, and 10\% of data to perform the experiments on three datasets, the results consistently validate the effectiveness and superiority of the proposed approach. In other words, we can obtain consistent results using only a small portion of the data. From the cost perspective, PiCO has the same complexity as the Chatbot Arena as the model amount scales up. Both tokens consumed will increase, but the human annotation cost of PiCO is significantly lower because it does not require any annotation.
>
> **Q3: While the three metrics you use are meaningful, you should also present results for Precision@1 and RBP@3.**
>
> **A**: Thanks for your valuable suggestion. We demonstrated the results of precision and RBP as follows.
>
> |RBP@|3|4|5|6|7|8|9|10|
> |-|-|-|-|-|-|-|-|-|
> |GPTScore(flan-t5-xxl)|0%|0%|0%|16.8%|22.0%|26.2%|29.6%|45.1%|
> |GPTScore(davinci-002)|0%|12.8%|31.2%|37.8%|37.8%|42.0%|50.6%|53.3%|
> |PandaLM|16.0%|36.0%|44.2%|63.5%|63.5%|63.5%|63.5%|66.2%|
> |PRD|28.8%|36.0%|67.2%|73.8%|73.8%|78.0%|81.3%|81.3%|
> |PRE|28.8%|48.8%|67.2%|73.8%|73.8%|78.0%|81.3%|81.3%|
> |**PiCO(ours)**|**28.8%**|**59.0%**|**67.2%**|**73.8%**|**73.8%**|**83.2%**|**83.2%**|**85.9%**|
>
> |Precision@|1|2|3|4|5|6|7|8|9|10|
> |-|-|-|-|-|-|-|-|-|-|-|
> |GPTScore(flan-t5-xxl)|0%|0%|0%|0%|0%|33.3%|42.9%|50.0%|55.6%|70.0%|
> |GPTScore(davinci-002)|0%|0%|0%|25.0%|60.0%|66.7%|57.1%|62.5%|77.8%|80.0%|
> |PandaLM|0%|50.0%|33.3%|50.0%|60.0%|83.3%|71.4%|62.5%|55.6%|60.0%|
> |PRD|0%|50.0%|66.7%|75.0%|80.0%|100.0%|85.7%|87.5%|88.9%|80.0%|
> |PRE|0%|50.0%|66.7%|100.0%|80.0%|100.0%|85.7%|87.5%|88.9%|80.0%|
> |**PiCO(ours)**|**0%**|**50.0%**|**66.7%**|**100.0%**|**100.0%**|**100.0%**|**85.7%**|**100.0%**|**100.0%**|**90.0%**|
>
> The results show that the proposed PiCO approach can achieve better precision and RBP performance in most cases. In addition, as suggested by Reviewer \#8Eov, **we also construct our experiments under more ranking metrics, _i.e._, Spearman's rank correlation coefficient $S(\uparrow)$, and Kendall's rank correlation coefficient $\tau(\uparrow)$**. These results in **global author rebuttal (Table R1)** once again validate the superiority of the proposed PiCO approach.
>
> **Q4: Fig 5 should be extended to nine graphs (3 metrics * 3 datasets). Discuss that not all "CIN = 1"  are created equal.**
>
> **A**: The nine graphs (3 metrics * 3 datasets) are demonstrated in **Figure R4 of rebuttal material**. The results show that the proposed PiCO approach can achieve better performance than PRE in most cases. In addition, "CIN = 1" can not be always created equal. Therefore, other ranking metrics, such as PEN, LIS, Spearman's, and Kendall's rank correlation coefficients, have been employed to help us validate the effectiveness of the proposed approach from other perspectives.
>
> **Q5: What is the benefit of a model evaluating itself? why not explicitly disallow it? w/o a full explanation, this detail unnecessarily complicates the story.**
>
> **A**: Thanks for your valuable suggestion. Evaluating itself or not will not have an impact on the final results. We will simplify the expression of this detail in our revised paper.
>
> **Q6: You repeatedly talk about ``higher-level LLM'' w/o defining the concept; is it based on the number of parameters (larger or higher)?**
>
> **A**: In this paper, "higher-level LLMs" indicate those LLMs with stronger capabilities or better performance. It is not entirely related to parameters, even though models with larger parameter quantities often have stronger capabilities.
>
> **Q7: Can PiCO (or an extension) also estimate the gap between the performance of two models that are adjacent in the final ranking?**
>
> **A**: Yes. As shown in **Figure R2 of the rebuttal material**, the learned each LLM's ability $w$ can better estimate the performance gap between different models. The results in Figure 4 also show that re-weighting by the learned weight $w$ can effectively alleviate the whole system evaluation bias, which can better reflect the ability of each model from another perspective.

---

> > ### Comment · Reviewer_e5pS · 2024-08-08
> > **Many thanks to the authors for the detailed answers. After considering both the other reviews and the rebuttals, I maintain the original rating.**
> >
> > Many thanks to the authors for the detailed answers. After considering both the other reviews and the rebuttals, I maintain the original rating.

---

### Official Review · Reviewer_8Eov · 2024-07-12

**Soundness:** 2
**Presentation:** 3
**Contribution:** 2
**Rating:** 4
**Confidence:** 4

**Summary:**

This paper studies how to estimate LLMs' performance ranking without human preference annotations. In particular, it proposes to leverage three metrics (PEN, CIN, LIS) to evaluate the estimation quality, gives an estimation mechanism that first asks a list of LLMs (called "reviewers") to rank pairwise answers to user questions independently, and then aggregates their ranking via a weighted sum approach. A consistency optimization determines the weights of each reviewer.

**Strengths:**

The problem of LLM evaluation without human annotations is critical in resource-limited applications. The most important and interesting contribution of this paper, in my opinion, is proposing the problem of estimating the performance rank of LLMs instead of any metric of an individual LLM. The paper also reveals an interesting assumption that better reviewers are expected to be better answer generators, which leads to their consistency optimization approach. Overall, the paper is well-written and easy to follow.

**Weaknesses:**

While I find the proposed problem interesting, there are still a few limitations, unfortunately.

***Unclear implication of ground truth ranking***: The technical part of the paper starts by introducing a ground truth ranking (equation (1)) without giving its physical meaning. It simply assumes "[...] alignment with human preferences", but it is not clear what human preferences mean in this context.

***Evaluation metric is strange***: One of my major concerns is on the choices of evaluation metric. All the three proposed metrics, PIN, CIN, LIS, in the authors' own words, seem originally used for time series comparison. However, the goal here is to compare rankings, not time series. Thus, it is unclear why we should not use the standard ranking comparison metrics, e.g., Spearman's rank correlation coefficient or Kendall rank correlation coefficient.

***Consistency optimization algorithm is not provided***: The core of the proposed ranking estimation method is the optimization problem (7). It does not seem to be a standard optimization problem, but I could not find (even a discussion on) any clue on how to solve it in this paper.

***An optimal solution to the consistency optimization formulation can be useless***: I find the following optimal solution to the problem (7): just set weight w to be 0 for all LLMs. It is an optimal solution as G and w are identical and thus the objective is always maximized.  However, this solution is undesired. I probably misunderstood something, but this seems to suggest the formulation is incorrect.

***Consistency optimization formulation seems brittle to query distribution biases***: Another problem with the formulation is that it seems brittle to data distribution bias. E.g., suppose M1 is indeed better than M2 for some query q. And let us replicate many copies of q in the dataset D. Then the grade G1 can be arbitrarily large. In other words, the grade of an LLM is proportional to the number of battles involving it in the dataset D, which should not be the case.

***Choices of LLMs for evaluation***: In line 547, the authors write "For our analysis, we meticulously selected 15 LLMs". What is the principle of the meticulous selection? Other than open-source and close-source, the selection is quite arbitrary. For example, I am quite surprised to not see GPT-4 and Claude included in the reviewer LLMs.

***Comparison with a simple baseline***: One simple baseline is to ask a powerful LLM(e.g., GPT-4, Cluade-3) to give a preference for each answer question pair, and then take the vote to determine the ranking. I would suggest to compare the proposed method with this simple baseline.

**Questions:**

***Unclear implication of ground truth ranking***: What is the physical meaning of the ground truth ranking? What does that imply?

***Evaluation metric is strange***: What is the motivation of using time series comparison metrics to compare rankings? How does the proposed method work under ranking comparison metrics?

***Consistency optimization algorithm is not provided***: How do you solve problem (7)?

***An optimal solution to the consistency optimization formulation can be useless***: How do you prevent the useless but optimal solution? Does that imply the formulation is incorrect?

***Query distribution and LLM judgment biases***: How could this formulation resist query distribution and LLM biases?

***Choices of LLMs for evaluation***: What is the principle of the meticulous selection? How does the proposed approach work if strong models such as GPT-4 and Claude are included in the reviewer LLMs?

***Comparison with a simple baseline***: How does the proposed method compare with the simple baseline?

**Limitations:**

No. The limitations are not well discussed.

---

> ### Author Rebuttal · Authors · 2024-08-07
>
> Thank you for your very detailed comments and suggestions for improvement. And we also appreciate your recognition of our **interesting idea and assumption**. Next, we will answer all your questions one by one.
>
> **Q1: Unclear implication of ground truth ranking: What is the physical meaning of the ground truth ranking? What does that imply?**
>
> **A**: The ground-truth ranking $\mathcal{R}^*$ means that the ranking by human annotation, like Chatbot Arena. It constructs anonymous battles between different LLMs and collects real human preferences to rank their ability. The physical meaning is the ranking that aligns with human preference. The goal of this paper is to learn a ranking $\hat{\mathcal{R}}$ in an unsupervised way that aligns with $\mathcal{R}^*$ as much as possible. We will discuss this more clearly in our revision.
>
> **Q2: Evaluation metric is strange: What is the motivation of using time series comparison metrics to compare rankings? How does the proposed method work under ranking comparison metrics?**
>
> **A**: Permutation Entropy (PEN) is widely used to evaluate the degree of confusion in an array, while Count Inversions (CIN) and Longest Increasing Subsequence (LIS) can reflect how far (or close) the array is from being sorted. We believe that PEN, CIN, and LIS can well assess the alignment between the learned ranking $\hat{\mathcal{R}}$ with ground-truth ranking $\mathcal{R}^*$ from multiple perspectives. Additionally, we appreciate your valuable suggestion that validates the proposed approach in more standard ranking comparison metrics, *i.e.*, Spearman's rank correlation coefficient $S(\uparrow)$, and Kendall's rank correlation coefficient $\tau(\uparrow)$. We conduct the experiments in these metrics, and the results are demonstrated in the **global author rebuttal (Table R1)**. It can be observed that the proposed PiCO approach also achieves the best performance under the new ranking metrics in all cases. These results once again validate that consistency optimization can learn a better ranking aligned with real human preference. We will add these results in our revision.
>
> **Q3: Consistency optimization algorithm is not provided: How do you solve the problem 7?**
>
> **A**: We solve the problem 7 in an iteration way. Firstly, each LLM’s capabilities $w$ are randomly initialized. Then, we calculate the score $G_j$ of each model $j$. Next, we compute the gradient $\Delta w = \frac{\partial\, \text{Consistency}(G, w)}{\partial w}$ and update $w=w+\Delta w$. In the next iteration process, we use the new $w$ to calculate the LLM's score and iteratively update $w$ until convergence. We will add these discussions in our revised paper.
>
> **Q4: An optimal solution to the consistency optimization formulation can be useless: How do you prevent the useless but optimal solution? Does that imply the formulation is incorrect?**
>
> **A**: In the consistency optimization formulation, we employ the Pearson correlation to measure the consistency between w and G, *i.e.*, $r = \frac{\sum_{i=1}^{n} (G_i - \overline{G})(w_i - \overline{w})}{\sqrt{\sum_{i=1}^{n} (G_i - \overline{G})^2} \sqrt{\sum_{i=1}^{n} (w_i - \overline{w})^2}}$. If $w$ and $G$ are equal to 0 for all LLMs, this formula will be meaningless. It is worth noting that the $w$ are randomly initialized to a value greater than 0 and less than 1. During the consistency optimization process, it is almost impossible for $w$ and $G$ to be optimized to be both 0. To validate it, we repeated the experiment with different seeds for 1000 times, and plotted the training loss curve and weight distribution in the **additional rebuttal material (Figure R1 and R2)**. The results show that the learning process is stable and the learned $w$ is convergence and greater than 0.
>
> **Q5: Query distribution and LLM judgment biases: How could this formulation resist query distribution and LLM biases?**
>
> **A**: In our experiments, we used the most popular crowdsourced datasets, *e.g.*, Chatbot Arena. It constructs anonymous battle platforms, where the user will ask a question $Q_i$ with two Chatbots simultaneously and rate their responses based on personal preferences. We removed those preference annotations and only used these human questions in our experiments. **In other words, the query distribution in our experiments is completely the same with the widely-used crowdsourced platforms. Most importantly, we believe that a good evaluation should also consider user query distribution.** Usually, we say Model A is better than Model B in terms of user experience, a possible reason is that Model A performs better in most common user queries, rather than uncommon problems. Therefore, we believe that the widely accepted evaluation method is to consider user queries, such as the most popular LLMs evaluation platforms (Chatbot Arena).
>
> **Q6: Choices of LLMs for evaluation: What is the principle of the meticulous selection? How does the proposed approach work if strong models such as GPT-4 and Claude are included in the reviewer LLMs?**
>
> **A**: The LLMs selection is based on the Chatbot Arena dataset [1], which includes 33K responses by 20 open-source and close-source LLMs. Among them, the APIs of GPT-4, Claude-v1, Claude-instant-v1, and PaLM-2 are too expensive, so we removed them from the reviewer queue. The official open-source FastChat code [2] has some issues in implementing the RWKV model, so we also remove it from the reviewer queue. Thus the remaining 15 models are used in our experiments.
>
> **Q7: Comparison with a simple baseline: How does the proposed method compare with the simple baseline?**
>
> **A**: We compare the Cluade-3 in our experiments, and the comparison results are also demonstrated in the **global author rebuttal (Table R1)**. Compared with Cluade-3, the PiCO can consistently achieve better performance in all cases.
>
> ---
> ### References
> [1] https://huggingface.co/datasets/lmsys/chatbot_arena_conversations
>
> [2] https://github.com/lm-sys/FastChat

---

> > ### Author Response · Authors · 2024-08-12
> >
> > Dear Reviewer 8Eov,
> >
> > We would like to once again thank you for your valuable time and constructive comments. This is a gentle reminder that we have diligently addressed your concerns and clarified any confusion. We have not yet heard back from you and would appreciate any further feedback you may have. If there are any concerns that you feel have not been fully addressed, we are eager to discuss them with you.
> >
> > Best regards!
> >
> > Authors

---

> ### Author Response · Authors · 2024-08-13
>
> Dear Reviewer 8Eov,
>
> We have provided detailed explanations and additional experimental results **(_e.g._ performance comparison in more ranking metrics and comparison with Claude (Table R1 of global author rebuttal))** to address all your concerns. With just **20 hours** left in the discussion period, we would appreciate it if you could review our responses and let us know if there are any final questions or concerns.  Your insights have been invaluable, and we thank you sincerely for your time and effort.
>
> Best regards!
>
> Authors

---

> > ### Comment · Reviewer_8Eov · 2024-08-13
> > **Thank you for the detailed response**
> >
> > Thank you for the detailed response. While the response does not fully address my concerns, I will keep my original score.

---

> > > ### Author Response · Authors · 2024-08-14
> > >
> > > May I ask which concerns have not been addressed? Can you provide more details? We are eager to discuss them with you.

---

### Official Review · Reviewer_Ezxz · 2024-07-13

**Soundness:** 2
**Presentation:** 2
**Contribution:** 1
**Rating:** 3
**Confidence:** 4

**Summary:**

In this paper, the authors propose a more reliable evaluation system to rank the abilities of different large language models (LLMs). Previous evaluation methods typically suffer from two main drawbacks: (1) benchmark data leakage and (2) cost-intensive and potentially biased human evaluations. To address these issues, the authors introduce an unsupervised evaluation mechanism to measure the abilities of LLMs and derive their rankings. The core idea of this mechanism is to first collect the answers from each LLM, then treat each LLM as a 'reviewer' to rate the quality of the other LLMs' answers, and finally optimize the internal agreement of the reviews among all LLMs. They also conduct experiments on three datasets to validate the effectiveness of their proposed mechanism.

**Strengths:**

1. Unsupervised Evaluation Method: The paper introduces PiCO (Peer Review in LLMs based on Consistency Optimization), a new unsupervised evaluation method that leverages peer-review mechanisms to measure the capabilities of LLMs automatically, particularly without human-annotated data. The unsupervised nature also makes it scalable and less subjectively biased.
2. Consistency Optimization Framework: The proposed approach includes a constrained optimization method based on the consistency assumption, which helps in re-ranking LLMs to align more closely with human preferences.
3. New Evaluation Metrics: The paper proposes three new metrics—Permutation Entropy (PEN), Count Inversions (CIN), and Longest Increasing Subsequence (LIS)—to evaluate the alignment of LLM rankings with human preferences. These metrics can further inspire future work.

**Weaknesses:**

1. Reliance on Consistency Assumption: The effectiveness of the method relies on the consistency assumption that higher-level LLMs can more accurately evaluate others. However, a natural concern is, "Does this assumption always hold true in practice?" I suggest the authors further discuss the applicability of their method.
2. Complexity of Implementation: The framework involves a complex review process, which requires substantial computational resources to support the LLMs' inference. Can you provide some details on the number of tokens consumed and a comparison of consumption with baseline methods?
3. Consideration of Multi-Agent Methods: Since the proposed method employs multiple LLMs, I think more recent and advanced evaluation methods based on multi-agent systems should also be considered in the experiments, such as AgentVerse.
4. Details of ELO: The ELO system is essential for the proposed mechanism. However, it is only mentioned in the appendix. I suggest the authors add more details about the background of ELO and how it is adopted in the proposed mechanism.

**Questions:**

Please see my above comments

**Limitations:**

Yes

---

> ### Author Rebuttal · Authors · 2024-08-06
>
> We appreciate your detailed comments. We would like to address your concerns below.
>
> **Q1: Reliance on Consistency Assumption: The effectiveness of the method relies on the consistency assumption. However, a natural concern is, "Does this assumption always hold true in practice?" I suggest the authors further discuss the applicability of their method.**
>
> **A**: The validation of the consistency assumption can be found in Section 3.4, and the results are displayed in Table 2. We compared with four baselines: "Forward Weight Voting", "Uniform Weight Voting", "Backward Weight Voting", and "Random Weight Voting + Consistency Optimization". Here we copy the results of Table 2 as follows,
>
> | | Chatbot Arena | | MT-Bench | | AlpacaEval | |
> |-|-|-|-|-|-|-|
> | Methods | PEN $(\downarrow)$ | CIN $(\downarrow)$ | PEN $(\downarrow)$ | CIN $(\downarrow)$ | PEN $(\downarrow)$ | CIN$(\downarrow)$|
> | Average Performance of Reviewer Queue | $1.49^{\pm0.28}$| $34.87^{\pm14.68}$| $1.49^{\pm0.26}$| $38.80^{\pm19.28}$| $1.50^{\pm0.23}$| $33.13^{\pm13.97}$|
> | Backward Weight Voting| $1.43^{\pm0.04}$| $25.00^{\pm0.00}$| $1.43^{\pm0.05}$| $26.00^{\pm0.00}$| $1.36^{\pm0.03}$| $24.00^{\pm0.00}$|
> | Uniform Weight Voting| $1.34^{\pm0.23}$| $22.00^{\pm0.00}$| $1.39^{\pm0.02}$| $24.00^{\pm0.00}$| $1.34^{\pm0.03}$| $22.00^{\pm0.00}$|
> | Forward Weight Voting | $1.32^{\pm0.03}$| $21.00^{\pm0.00}$| $1.33^{\pm0.03}$| $23.00^{\pm0.00}$| $1.30^{\pm0.05}$| $21.00^{\pm0.00}$|
> | **Random Weight + Consistency Optimization**| **1.17**$^{\pm0.06}$ | **17.50**$^{\pm0.50}$| **1.20**$^{\pm0.08}$| **18.00**$^{\pm1.22}$| **1.21**$^{\pm0.04}$| **19.00**$^{\pm0.00}$|
> ----
>
> From Table 2, we can summarize the following two conclusions:
> - Forward Weight Voting consistently outperforms Uniform and Backward Weight Voting in all cases, while Backward Weight Voting yields the worst results. This suggests that assigning larger weights to models with stronger capabilities results in better alignment, supporting the Consistency Assumption.
> - Starting with randomly initialized weights, our **_consistency optimization_** algorithm further improves performance across all cases. This demonstrates that adjusting weights based on our algorithm enhances the reliability of evaluations, further validating the Consistency Assumption.
>
> Additionally, as recommended by Reviewer \#8Eov, we conducted experiments using additional ranking metrics, such as Spearman's and Kendall's rank correlation coefficients,
>
> | | Chatbot Arena | | MT-Bench | | AlpacaEval | |
> |-|-|-|-|-|-|-|
> | Methods | $S (\uparrow)$ | $\tau (\uparrow)$ | $S (\uparrow)$ | $\tau (\uparrow)$ | $S (\uparrow)$ | $\tau (\uparrow)$|
> | Backward Weight Voting| $0.70^{\pm0.00}$| $0.50^{\pm0.00}$| $0.72^{\pm0.00}$| $0.52^{\pm0.00}$| $0.69^{\pm0.00}$| $0.50^{\pm0.00}$|
> | Uniform Weight Voting| $0.74^{\pm0.00}$| $0.54^{\pm0.00}$| $0.80^{\pm0.00}$| $0.58^{\pm0.00}$| $0.77^{\pm0.00}$| $0.58^{\pm0.00}$|
> | Forward Weight Voting | $0.75^{\pm0.00}$| $0.56^{\pm0.00}$| $0.82^{\pm0.00}$| $0.59^{\pm0.00}$| $0.79^{\pm0.00}$| $0.60^{\pm0.00}$|
> | **Random Weight + Consistency Optimization**| **0.90**$^{\pm0.00}$ | **0.77**$^{\pm0.00}$| **0.89**$^{\pm0.01}$| **0.72**$^{\pm0.01}$| **0.84**$^{\pm0.00}$| **0.68**$^{\pm0.00}$|
> ----
>
> The results from these metrics are consistent with our previous findings, further confirming the validity of the Consistency Assumption in the LLM peer-review process.
>
> **Q2: Complexity of Implementation: The framework involves a complex review process, which requires substantial computational resources to support the LLMs' inference. Can you provide some details on the number of tokens consumed and a comparison of consumption with baseline methods?**
>
> **A**: The review process will not result in excessive token consumption, as each reviewer will only evaluate a small number of question-answer pairs (line 142-143). The comparison of tokens consumed in Chatbot Arena dataset is demonstrated as follows.
>
> | Methods | Input Token | Output Token  | Annotation Cost |
> | - | - | - | - |
> | Chatbot Arena Platforms (Fully Human Evaluation) | ~7,500k | ~10,944k  | 32k |
> | GPTScore (flan-t5-xxl) | ~22,882k | ~12,260k  | 0 |
> | GPTScore (davinci-002) | ~22,882k | ~12,260k  | 0 |
> | PandaLM | ~22,882k | ~10,355k  | 0 |
> | PRD | ~25,087k | ~10,935k  | 0 |
> | PRE (Partially Human Evaluation) | ~24,120k | ~11,115k  | 7k |
> | **PiCO (ours)** | ~23,823k | ~11,685k  | 0 |
> ---
>
> The proposed PiCO approach has a similar token consumed with other baselines while achieving better evaluation performance. Although Chatbot Arena has a smaller token consumption, it requires 33k human annotations, while **PiCO does not require any human annotations**.
>
> **Q3: Consideration of Multi-Agent Methods: Since the proposed method employs multiple LLMs, I think more recent and advanced evaluation methods based on multi-agent systems should also be considered in the experiments, such as AgentVerse.**
>
> **A**: To the best of our knowledge, the AgentVerse is designed to construct multi-agent systems to better solve tasks and do simulations in real applications. It seems not related to our work about LLMs' unsupervised evaluation. We appreciate your suggestion on considering multi-agent methods. In fact, the compared methods including Majority Voting, Rating Voting, PRD, and PRE are already performed under the multi-agent framework. We will add these discussions about AgentVerse in our revised paper.
>
> **Q4: Details of ELO: The ELO system is essential for the proposed mechanism. However, it is only mentioned in the appendix. I suggest the authors add more details about the background of ELO and how it is adopted in the proposed mechanism.**
>
> **A**: We appreciate your valuable suggestion. The ELO mechanism is employed to prevent a model from continuously defeating weaker opponents to achieve a higher ranking. We will move the details of the ELO from the appendix to the main text, and attach more discussions in our revision.

---

> > ### Author Response · Authors · 2024-08-12
> >
> > Dear Reviewer Ezxz,
> >
> > We would like to once again thank you for your valuable time and constructive comments. This is a gentle reminder that we have diligently addressed your concerns and clarified any confusion. We have not yet heard back from you and would appreciate any further feedback you may have. If there are any concerns that you feel have not been fully addressed, we are eager to discuss them with you.
> >
> > Best regards!
> >
> > Authors

---

> ### Author Response · Authors · 2024-08-13
>
> Dear Reviewer Ezxz,
>
> We have provided detailed explanations and additional experimental results **(e.g. further validation of the consistency assumptions)** to address all your concerns. With just **20 hours** left in the discussion period, we would appreciate it if you could review our responses and let us know if there are any final questions or concerns.  Your insights have been invaluable, and we thank you sincerely for your time and effort.
>
> Best regards!
>
> Authors

---

### Official Review · Reviewer_mhTC · 2024-07-17

**Soundness:** 3
**Presentation:** 3
**Contribution:** 2
**Rating:** 7
**Confidence:** 4

**Summary:**

The highlight of the work is the proposed method of evaluating Large Language Models without relying on human feedback.

**Strengths:**

- The proposed evaluation method is a novel attempt of automating the LLM improvement process. Such method worth further exploration. It could be adapted to many of the LLMs and potentially bring us more insights.
- By eliminating the involvement of human, the proposed evaluation method limits the bias brought by human labelers. The observations presented are also interesting as LLMs can sometimes surprise us.
- Great presentation and visualization.

**Weaknesses:**

Some of the equations and notations in the paper seems unnecessarily complicated, which can be reorganized when polishing.

**Questions:**

1. How do you define unlabeled questions in line76, as in to what extend is a question unlabeled?
2. How do you differentiate high-level from low-level LLMs? Are they interchangeable or more of a dynamic definition while evaluating on different tasks?

**Limitations:**

The idea is straightforward and make sense to me. But LLMs can be trained to bypass such systems, which may lead to potential fairness or security problems.

---

> ### Author Rebuttal · Authors · 2024-08-06
>
> Thank you for your recognition of our **interesting observations, good presentation, and great visualization**. And now we will explain and discuss the questions you raised.
>
> **Q1: How do you define unlabeled questions in line76, as in to what extend is a question unlabeled?**
>
> **A**: The "unlabeled questions" indicate the questions without model responses and human preference annotations. For example, for a question" ($Q_i$): who is the founder of Apple? The response ($A_i^j$) of model $j$: Steve Jobs. The response ($A_i^k$) of model $k$: Bill Gates. Human  Annotation ($A_i^j>A_i^k$): model $j$ is more correct than model $k$. Thus the "unlabeled question" only indicate the $Q_i$.
>
> **Q2: How do you differentiate high-level from low-level LLMs? Are they interchangeable or more of a dynamic definition while evaluating on different tasks?**
>
> **A**: The ability levels of different LLMs are learned by our **_consistency optimization_**, which aims to maximize the
> consistency of each LLM’s capabilities and scores. It is thus a dynamic process when evaluating different tasks.
>
> **Q3: Limitation on bypassing evaluation systems, results in potential fairness and security problems.**
>
> **A**: We appreciate your valuable suggestion. LLMs may indeed train to bypass the evaluation system and deceive reviewers to achieve higher rankings.  We will discuss this limitation in our revised paper.

---

> > ### Author Response · Authors · 2024-08-12
> >
> > Dear Reviewer mhTC,
> >
> > We would like to once again thank you for your valuable time and constructive comments. This is a gentle reminder that we have diligently addressed your concerns and clarified any confusion. We have not yet heard back from you and would appreciate any further feedback you may have. If there are any concerns that you feel have not been fully addressed, we are eager to discuss them with you.
> >
> > Best regards!
> >
> > Authors

---

### Author Rebuttal · Authors · 2024-08-07

## To All Reviewers
We thank all reviewers for their thoughtful and constructive comments on our works. We have responded to all reviewers' comments and uploaded an additional rebuttal material (PDF). The PDF includes four figures about, learning stability (Figure R1 for Review \#8Eov), weight non-zero verification (Figure R2 for Review \#8Eov), dynamic threshold (Figure R3 for Review \#e5pS), and nine graphs (Figure R4 for Review \#e5pS).

Moreover, **we particularly appreciate the valuable suggestions by Reviewers \#8Eov and \#e5pS that validate the proposed approach in more standard ranking comparison metrics.** We thus conduct the experiments in Spearman’s Rank $(S\uparrow)$ and Kendall’s Tau $(\tau\uparrow)$ Correlation Coefficients as shown in the flowing Table R1. And we will add these results in our revision.

---

#### **Table R1: Comparison of all methods on three datasets under data percentage of 1, 0.7, 0.4, and 0.1. Highlighting top values in bold font. Higher $S$ (Spearman’s Rank Correlation Coefficient) and $\tau$ (Kendall’s Tau) indicate better performance.**
---
| Methods |*| Chatbot|Arena  | * | * | MT |Bench| * | * |Alpaca|Eval| * |
|-|-|-|-|-|-|-|-|-|-|-|-|-|
|||||$S$ $(\uparrow)$|**(Spearman’s**|**Rank**|**Correlation**|**Coefficient)**|||||
|Data Percentage | 1| 0.7| 0.4| 0.1| 1| 0.7| 0.4| 0.1| 1| 0.7| 0.4| 0.1|
| Majority Voting| $0.76^{\pm0.00}$ | $0.75^{\pm0.01}$ | $0.73^{\pm0.03}$ | $0.67^{\pm0.08}$ | $0.73^{\pm0.00}$ | $0.77^{\pm0.01}$ | $0.75^{\pm0.01}$ | $0.74^{\pm0.04}$ | $0.80^{\pm0.00}$ | $0.79^{\pm0.01}$ | $0.78^{\pm0.01}$ | $0.80^{\pm0.03}$ |
| Rating Voting|$0.74^{\pm0.00}$ |$0.72^{\pm0.02}$ |$0.71^{\pm0.02}$|$0.66^{\pm0.03}$|$0.80^{\pm0.00}$ |$0.78^{\pm0.02}$ |$0.74^{\pm0.03}$|$0.71^{\pm0.04}$|$0.77^{\pm0.00}$ |$0.77^{\pm0.01}$ |$0.78^{\pm0.01}$|$0.78^{\pm0.01}$|
| GPTScore(flan-t5-xxl)| $-0.09^{\pm0.00}$ | $-0.09^{\pm0.01}$ | $-0.12^{\pm0.02}$ | $-0.19^{\pm0.06}$ | $0.05^{\pm0.00}$ | $0.01^{\pm0.07}$ | $0.04^{\pm0.09}$ | $-0.19^{\pm0.06}$ | $0.34^{\pm0.00}$ | $0.34^{\pm0.00}$ | $0.34^{\pm0.01}$ | $-0.19^{\pm0.06}$ |
| GPTScore(davinci-002) |$0.15^{\pm0.00}$ |$0.13^{\pm0.02}$ |$-0.02^{\pm0.14}$|$-0.19^{\pm0.06}$|$0.52^{\pm0.00}$ |$0.42^{\pm0.05}$ |$0.45^{\pm0.05}$|$-0.19^{\pm0.06}$|$0.76^{\pm0.00}$ |$0.77^{\pm0.07}$ |$0.75^{\pm0.06}$|$-0.19^{\pm0.06}$ |
| PandaLM |$0.43^{\pm0.00}$ |$0.44^{\pm0.03}$ |$0.44^{\pm0.10}$| $0.51^{\pm0.15}$|$0.50^{\pm0.00}$ |$0.50^{\pm0.08}$ |$0.52^{\pm0.17}$|$0.50^{\pm0.22}$|$0.57^{\pm0.00}$ |$0.55^{\pm0.01}$ |$0.48^{\pm0.08}$|$0.45^{\pm0.19}$ |
| PRD |$0.84^{\pm0.00}$ | $0.84^{\pm0.00}$|$0.82^{\pm0.03}$|$0.80^{\pm0.04}$|$0.86^{\pm0.00}$ |$0.84^{\pm0.03}$ |$0.81^{\pm0.03}$|$0.77^{\pm0.08}$|$0.81^{\pm0.00}$ |$0.81^{\pm0.01}$ |$0.81^{\pm0.02}$|$0.82^{\pm0.03}$ |
| PRE |$0.86^{\pm0.00}$ |$0.86^{\pm0.01}$ |$0.86^{\pm0.01}$|$0.84^{\pm0.02}$|$0.86^{\pm0.00}$ |$0.84^{\pm0.03}$ |$0.82^{\pm0.04}$|$0.75^{\pm0.04}$|$0.83^{\pm0.00}$ |$0.81^{\pm0.01}$ |$0.83^{\pm0.02}$|$0.83^{\pm0.03}$ |
| Claude-3 **(R\#8Eov)** |$0.90^{\pm0.01}$ |$0.88^{\pm0.03}$ |$0.87^{\pm0.04}$ |$0.80^{\pm0.04}$|$0.85^{\pm0.06}$ |$0.82^{\pm0.08}$ |$0.80^{\pm0.07}$|$0.62^{\pm0.14}$|$0.79^{\pm0.03}$ |$0.78^{\pm0.02}$ |$0.75^{\pm0.04}$ |$0.79^{\pm0.06}$ |
| **PiCO (Ours)** | **0.90**$^{\pm0.00}$ | **0.89**$^{\pm0.01}$ | **0.89**$^{\pm0.01}$ | **0.86**$^{\pm0.05}$ | **0.89**$^{\pm0.01}$ | **0.89**$^{\pm0.01}$ | **0.84**$^{\pm0.11}$ | **0.78**$^{\pm0.21}$ | **0.84**$^{\pm0.00}$ | **0.83**$^{\pm0.03}$ | **0.85**$^{\pm0.01}$ | **0.84**$^{\pm0.02}$ |
|||||$\tau$ $(\uparrow)$|**(Kendall’s**|**Rank**|**Correlation**|**Coefficient)**|||||
|Majority Voting| $0.58^{\pm0.00}$ | $0.56^{\pm0.02}$ | $0.52^{\pm0.05}$| $0.45^{\pm0.09}$| $0.56^{\pm0.00}$ | $0.61^{\pm0.02}$ | $0.60^{\pm0.02}$| $0.55^{\pm0.05}$| $0.62^{\pm0.00}$ | $0.60^{\pm0.02}$ | $0.58^{\pm0.02}$| $0.62^{\pm0.04}$|
|Rating Voting| $0.54^{\pm0.00}$ | $0.53^{\pm0.02}$ | $0.52^{\pm0.02}$| $0.47^{\pm0.02}$| $0.58^{\pm0.00}$ | $0.57^{\pm0.02}$ | $0.54^{\pm0.01}$| $0.53^{\pm0.03}$| $0.58^{\pm0.00}$ | $0.57^{\pm0.01}$ | $0.57^{\pm0.02}$| $0.59^{\pm0.02}$|
|GPTScore(flan-t5-xxl) | $-0.06^{\pm0.00}$ |$-0.06^{\pm0.02}$ | $-0.09^{\pm0.02}$| $-0.13^{\pm0.07}$| $-0.05^{\pm0.00}$ | $-0.07^{\pm0.05}$ | $-0.02^{\pm0.06}$| $-0.13^{\pm0.07}$ | $0.25^{\pm0.00}$ | $0.26^{\pm0.01}$ | $0.26^{\pm0.01}$| $-0.13^{\pm0.07}$ |
|GPTScore(davinci-002) | $0.20^{\pm0.00}$ | $0.13^{\pm0.02}$ | $0.03^{\pm0.11}$| $-0.13^{\pm0.07}$ | $0.36^{\pm0.00}$ | $0.30^{\pm0.05}$ | $0.31^{\pm0.05}$| $-0.13^{\pm0.07}$ | $0.60^{\pm0.00}$ | $0.61^{\pm0.05}$ | $0.59^{\pm0.08}$| $-0.13^{\pm0.07}$ |
|PandaLM | $0.30^{\pm0.00}$ | $0.31^{\pm0.03}$ | $0.31^{\pm0.07}$| $0.38^{\pm0.10}$| $0.39^{\pm0.00}$ | $0.37^{\pm0.06}$ | $0.40^{\pm0.12}$| $0.36^{\pm0.17}$| $0.41^{\pm0.00}$ | $0.39^{\pm0.02}$ | $0.32^{\pm0.05}$| $0.33^{\pm0.12}$|
|PRD | $0.68^{\pm0.00}$ | $0.69^{\pm0.01}$ | $0.67^{\pm0.03}$| $0.63^{\pm0.06}$| $0.68^{\pm0.00}$ | $0.66^{\pm0.02}$ | $0.63^{\pm0.03}$| $0.60^{\pm0.08}$| $0.64^{\pm0.00}$ | $0.63^{\pm0.03}$ | $0.63^{\pm0.02}$| $0.64^{\pm0.05}$|
|PRE | $0.71^{\pm0.00}$ | $0.73^{\pm0.02}$ | $0.72^{\pm0.02}$| $0.69^{\pm0.04}$| $0.68^{\pm0.00}$ | $0.68^{\pm0.02}$ | $0.65^{\pm0.02}$| $0.61^{\pm0.04}$| $0.64^{\pm0.00}$ | $0.66^{\pm0.01}$ | $0.66^{\pm0.03}$| $0.65^{\pm0.03}$|
|Claude-3 **(R\#8Eov)** | $0.76^{\pm0.04}$| $0.72^{\pm0.05}$ | $0.70^{\pm0.07}$ | $0.60^{\pm0.05}$| $0.67^{\pm0.07}$ | $0.66^{\pm0.11}$ | $0.61^{\pm0.10}$ | $0.50^{\pm0.13}$ | $0.64^{\pm0.06}$ | $0.61^{\pm0.04}$ | $0.66^{\pm0.06}$ | $0.59^{\pm0.06}$|
| **PiCO (Ours)** | **0.77**$^{\pm0.00}$ | **0.76**$^{\pm0.01}$ | **0.77**$^{\pm0.02}$ | **0.71**$^{\pm0.07}$ | **0.72**$^{\pm0.01}$ | **0.72**$^{\pm0.03}$ | **0.70**$^{\pm0.12}$ | **0.62**$^{\pm0.20}$ | **0.68**$^{\pm0.00}$ | **0.66**$^{\pm0.04}$ | **0.67**$^{\pm0.02}$ | **0.67**$^{\pm0.01}$ |

---

### Decision · Program_Chairs · 2024-09-25

**Decision:**

Reject

**Comment:**

This paper addresses the important issue of evaluating LLMs without relying on human feedback.

The reviewers acknowledged the novelty and potential of the proposed method, but also raised a number of more serious concerns.

These concerns include clarifying the presentation (including writing, mathematical notation, equations, etc.), providing more evidence in support of the underlying assumptions (related to the existence of larger LLMs that can rate smaller ones consistently well), the computational cost of the approach, and fundamental questions about the evaluation methodology (why not consider more standard rank-based metrics?)

This is promising work and the authors are strongly encouraged to carefully consider the reviewer feedback when preparing an updated version of their paper to improve the clarity and to strengthen the arguments in favor of the proposed method.